# Coarse woody debris decomposition assessment tool: Model development and sensitivity analysis

**Zhaohua Dai**[1,2]*, **Carl C. Trettin**[1], **Andrew J. Burton**[2], **Martin F. Jurgensen**[2], **Deborah S. Page-Dumroese**[3], **Brian T. Forschler**[4], **Jonathan S. Schilling**[5], **Daniel L. Lindner**[6]

**1** Center for Forested Watershed Research, USDA Forest Service, Cordesville, South Carolina, United States of America, **2** College of Forest Resources and Environmental Science, Michigan Technological University, Houghton, Michigan, United States of America, **3** Rocky Mountain Research Station, USDA Forest Service, Moscow, Idaho, United States of America, **4** Department of Entomology, University of Georgia, Athens, Georgia, United States of America, **5** Plant & Microbial Biology, Itasca Biological Station & Laboratories, University of Minnesota, Saint Paul, Minnesota, United States of America, **6** Northern Research Station, USDA Forest Service, Madison, Wisconsin, United States of America

* zhaohuad@mtu.edu

**Data Availability Statement:** The main data used to parameterize the model were obtained from Daymet database (see reference: Thornton et al, 2016; doi: 10.3334/ORNLDAAC/1328). The

## Abstract

Coarse woody debris (CWD) is an important component in forests, hosting a variety of organisms that have critical roles in nutrient cycling and carbon (C) storage. We developed a process-based model using literature, field observations, and expert knowledge to assess woody debris decomposition in forests and the movement of wood C into the soil and atmosphere. The sensitivity analysis was conducted against the primary ecological drivers (wood properties and ambient conditions) used as model inputs. The analysis used eighty-nine climate datasets from North America, from tropical (14.2˚ N) to boreal (65.0˚ N) zones, with large ranges in annual mean temperature (26.5˚C in tropical to -11.8˚C in boreal), annual precipitation (6,143 to 181 mm), annual snowfall (0 to 612 kg m$^{-2}$), and altitude (3 to 2,824 m above mean see level). The sensitivity analysis showed that CWD decomposition was strongly affected by climate, geographical location and altitude, which together regulate the activity of both microbial and invertebrate wood-decomposers. CWD decomposition rate increased with increments in temperature and precipitation, but decreased with increases in latitude and altitude. CWD decomposition was also sensitive to wood size, density, position (standing vs downed), and tree species. The sensitivity analysis showed that fungi are the most important decomposers of woody debris, accounting for over 50% mass loss in nearly all climatic zones in North America. The model includes invertebrate decomposers, focusing mostly on termites, which can have an important role in CWD decomposition in tropical and some subtropical regions. The role of termites in woody debris decomposition varied widely, between 0 and 40%, from temperate areas to tropical regions. Woody debris decomposition rates simulated for eighty-nine locations in North America were within the published range of woody debris decomposition rates for regions in northern hemisphere from 1.6˚ N to 68.3˚ N and in Australia.

coordinates of the eighty-nine sites for obtaining climate data to analyze the model sensitivity are in S1 Table. S1, S2, S4 and S5 Tables have been added to let readers and model users can see the data we used, enabling them to also test the model using those data. The climate, coordinates and altitude of corresponding locations were from Daymet database used to analyze the model sensitivity, as long as the data are downloaded from Daymet and used to parameterize the model, the results from the model outputs can be used to compare with the results shown in this manuscript. The data in Table 1, S1 and S2 Tables are needed if anyone who wants to test this model. Other data used to parameterize the model are in this manuscript. S4A and S4B Table are those data collected for the comparison of modeling results and those from other studies.

**Funding:** Support of the FWDE and the model development has been provided by U.S. Dept. of Energy (DOE, USA), grant number DE-SC0016235 to CCT and the National Science Foundation (NSF, USA), grant number DEB 1754603 to AJB and grant number DEB 1754616 to JSS. The funders had no role in study design, data collection and analysis, decision to publish, or preparation of the manuscript.

**Competing interests:** The authors have declared that no competing interests exist.

# Introduction

Coarse woody debris (CWD) is an important component of forest carbon (C) pools [1, 2], and is structurally and functionally important for forest ecosystems [3, 4]. CWD provides habitats for a large variety of organisms [2], influences the potential risk of wild fires [5, 6], and plays important roles in C and nutrient cycling in forest ecosystems [7–12]. CWD is also important for mitigating climate change due to its size and relatively slow decomposition (wood mass loss to various decomposition processes, including biological consumption, fragmentation, and chemical dissolution) [13, 14]. Accordingly, assessing decomposition of CWD in forests is fundamental to understanding the importance of CWD in forest C cycling under changing climate and in response to forest management.

CWD decomposition can be divided into three processes: 1) chemical dissolution that occurs when water/rain acts as a solvent dissolving soluble materials in the wood [2]; 2) physical processes that include leaching of soluble material, wood fragmentation from seasonal and diurnal temperature differences, and external forces such as wind and water; and 3) biological-decomposition from the combination of bacterial, fungal and invertebrate organisms that dominate wood decomposition processes in nearly all ecosystems [2, 15–17].

Fungi have a key role in CWD decomposition in various eco-environments, with lignin removed or altered by both white-rot and brown-rot fungi [18]. Soft-rot fungi can decompose cellulose as brown-rot fungi do [2], however, they can tolerate high moisture, poor aeration, and low temperature [19–23], making them important for wood decomposition in very wet or extremely cold areas [24].

Decomposition of fine woody debris has been widely studied to assess its role in nutrient cycling [13], but less is known regarding the impact of CWD decomposition on soil nutrient pools due to its slow decomposition [25], which makes it difficult to assess CWD decomposition over a long time period under changing environmental conditions.

There are many factors that can influence CWD decomposition, including CWD properties, such as tree species, wood density, wood size, wood position (standing vs. down) [26], the type and species of wood-decomposers, and various ambient conditions, especially temperature and precipitation. Consequently, wood decomposition is a complicated process and difficult to measure, and unsurprisingly, different empirical decomposition models have been suggested for estimating wood mass loss under different eco-environmental conditions, including exponential (either single or multiple) and more complex decomposition models [7, 26–29]. Although empirical decomposition models are easy to use, they are mainly limited by the need to obtain reliable decomposition equations for specific study sites or from certain regions. However, mechanistic models are developed from expert knowledge, long-term experiences, and multiple field observations, which can give much better estimates of wood decomposition across a wide range of forest and climate conditions.

There are several computer models used to estimate CWD decomposition. For example, Yin [30] developed a computer model based on the methodology suggested by Agren and Bosatta [31] to analyze C and nitrogen dynamics in forest soils. Yasso, a process-based soil C model with a woody litter decomposition subroutine, was used to assess woody litter decomposition, and the results were consistent with the litterbag data from Canada [32, 33]. On the basis of these two computer models, Zell et al. [34] developed a computer model used to reanalyze woody debris decomposition using the same data as Yin [30]. Although these models are not completely process-based, they do give better estimations of CWD decomposition than simple empirical models.

Because CWD decomposition in a forest is controlled by complex ecological drivers that interact with other C pools and environmental fluxes, it is necessary to use a mechanistic

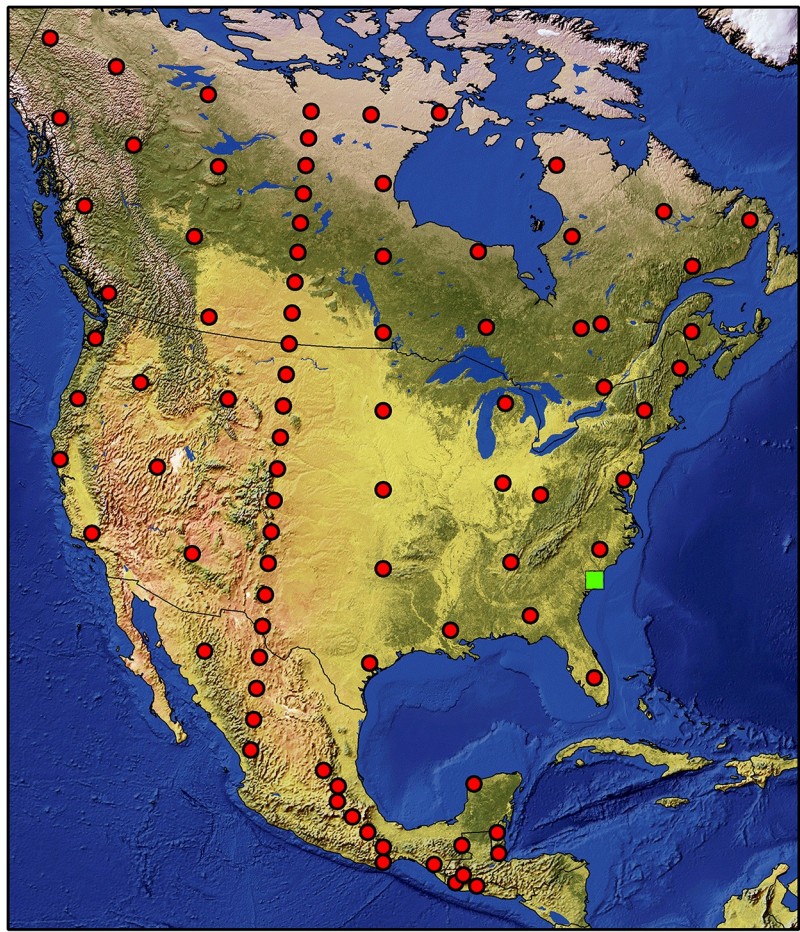

**Fig 1. Eighty-nine sites in North America used for model sensitivity analysis.** This figure was produced with the data from Natural Earth. Free vector and raster map data @ naturalearthdata.com; the green square is the location of Santee Experimental Forest.

model that reflects inherent biogeochemical processes. Accordingly, we have developed a process-based model CWDDAT (Coarse Woody Debris Decomposition Assessment Tool) to simulate CWD decomposition in forests by targeting CWD properties and ecological drivers, each of which affect CWD decomposition processes and the transfer of wood C to the atmosphere and soil by different biological communities.

In this paper we present the main equations used in the CWDDAT for modelling CWD decomposition in forests and results from a sensitivity analysis using model inputs of the eco-environmental conditions that impact CWD decomposition. Accordingly, the sensitivity analysis determined whether wood-decomposers are sensitive to changes in ecological drivers, such as wood properties and ambient conditions, and whether the computer model is stable over a range of climatic conditions. Eighty-nine North American climate datasets (Fig 1, S1 and S2 Tables) were obtained from the Daymet database [35] for locations ranging from 14˚ to 65˚ N latitude and 58˚ to 139˚ W longitude, and between 3 and 2,824 m above mean sea level. The sensitivity analysis also included different tree species groups (softwood and hardwood), size classes, wood density, and position of CWD (standing and downed).

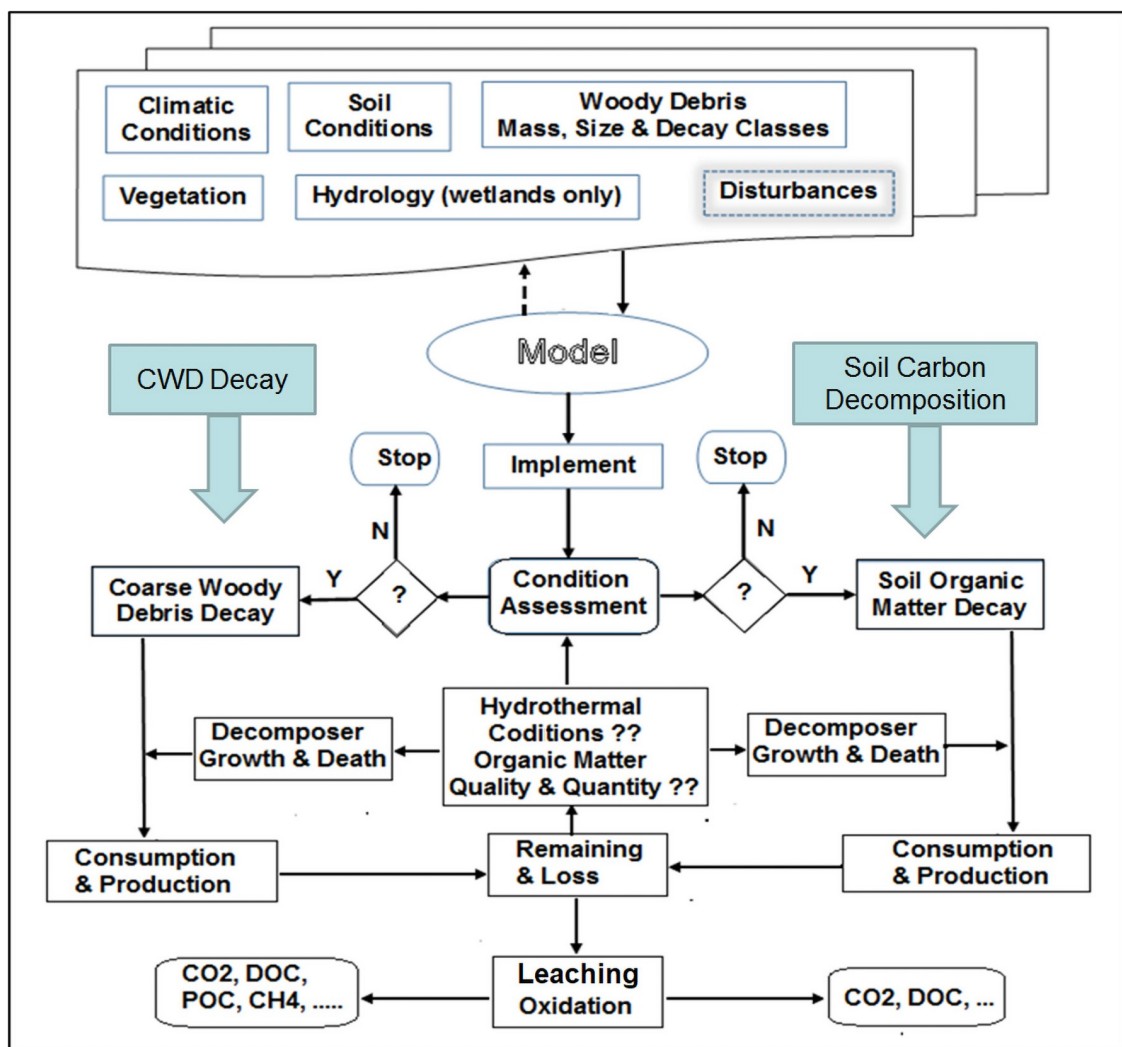

**Fig 2. Framework for modelling coarse woody debris decomposition simulation performance.** If CWD exists based on "Condition-Assessment", the model implements CWD decomposition, otherwise, the model will stop. When fragmentation and/or leaching occur, the model will assess carbon from CWD decomposition in soil dynamics, including decomposition, leaching and loss. The solid arrows are used to show the model performance directions: "N" means that the conditions are not satisfied for the model continuing to run; otherwise, the model will continue to run. If there is CWD and all conditions are good for one decomposer or all decomposers, the corresponding decay functions are called. The soil functions will be called whether or not there are fragments from CWD, to assess daily changes in soil moisture, temperature, soil organic carbon, and biological activity in soils.

## Modelling methods

### Model framework

The CWDDAT model was developed from coarse woody debris decomposition studies in the literature [2, 26, 36–38]. It is a process-based tool used to simulate CWD decomposition in forests (Fig 2) with a consideration that CWD decomposition processes are influenced by wood properties, decomposer community, and ambient conditions, including climate, soil properties, geographical location, altitude, and land cover. Biological, physical, and chemical processes simulated in CWDDAT include: biological action of fungi, termites, and other invertebrates with bacterial synergy; physical fragmentation and subsequent leaching; and temperature-dependent chemical reactions with precipitation.

## CWD decomposition

**Fragmentation.** Fragmentation is an important component of CWD decomposition because it reduces the original log into smaller pieces, which increases wood surface area [2]. It is a physical process, whereby external forces, including gravity, shrink-swell due to diurnal temperature changes, water scouring, freeze-thaw, and animal foraging, breaks the wood into smaller pieces. Accordingly, fragmentation was estimated as

$$FGM = \sum f_i \times m_d \tag{1}$$

where $FGM$ is the mass of fragments, kg; $f_i$ is the coefficient of the factor $i$; $m_d$ is the available mass for possible fragmentation at time $d$ (day), which can change with time and ecological conditions. For example, the loss of bark and branches from a log due to gravity can be estimated. The $m_d$ at time $d$ (day) was calculated

$$m_d = M_d \times (T_d + R_d) \times y \tag{2}$$

where $M_d$ is the largest value of available mass (kg) that is possibly lost at time $d$; $m_d$ is the actual available loss to gravity at time $d$; $T_d$ and $R_d$ are the coefficients of the temperature difference and ice, respectively, and $(T_d + R_d) \leq 1$; $y$ is a non-linear time function, $y \in (0, 1)$. Those fragments that separate from CWD are kept on the forest floor and subsequently (see Biological Process below).

**Biological processes.** Four principal biological agents were used to describe the decomposition processes: fungi, termites, beetles [39], and bacteria [2]. Fungi are the principal agents for decomposing CWD [24, 34, 40, 41], and their role in CWD decomposition was assessed by Eq (3):

$$MFungi = \sum_1^n B_i \times C_i \tag{3}$$

where $MFungi$ is CWD mass loss due to the decomposition by fungal group $i$ (mg day$^{-1}$); $B_i$ is the fungal biomass for group $i$; $C_i$ is the fungal respiration coefficient, 0.0807 [42]. The fungal biomass was calculated by Eq (4):

$$B_i = M_i \times P_i \times f_{ti} \times ff_i \tag{4}$$

where $M_i$ is daily available C for fungal group $i$ (kg day$^{-1}$); $P_i$ is fungal growth potential coefficient for group $i$ [42], $P \in [0, 1]$; $ff_i$ is the moisture coefficient of fungal group $i$; $f_{ti}$ is the coefficient for temperature. The ranges of temperature and moisture among the fungal groups are different. The temperature range is between 0 and 45˚C with an optimal temperature of 25˚C [2, 19] for white and brown-rot fungi, but between -45 and 45˚C with an optimal temperature of 25˚C for soft-rot fungi because soft-rot fungi can survive in colder and wetter ecological environments than white and brown-rot fungi. $ff_i$ is a moisture function, i.e., $ff_i = 1 - (\sin(|1 - S_a \div S_{ci}|))^{0.2}$ when $S_{ci} \neq S_a$, $ff_i = 0.5$ when $S_{ci} = S_a$, where $S_{ci}$ is the optimal moisture for fungal group $i$; $S_a$ is the actual moisture. Each coefficient is between 0.0 and 0.5 for each fungal group, but $\Sigma ff_i = 1$. The range of moisture content (fraction) is between 0.3 and 0.95 with an optimum 0.65 for brown-rot fungi, 0.3 and 1.0 with the optimum 0.75 for white-rot fungi, and 0.3 and 1.1 with the optimum 0.99 for soft-rot fungi based on the ranges reported by Harmon et al. [2] and Thybring et al. [43].

CWD decomposes much faster in areas with termites. The CWD mass loss to attributable to termite consumption was calculated as described in Eq (5):

$$MTermite = B \times C \tag{5}$$

where *MTermite* is the daily CWD consumed by termites (mg day$^{-1}$); *B* is the population of termites on the CWD; *C* is the termite consumption coefficient (0.015–0.2 mg per termite per day) [44, 45]. The value of *C* is species-based because of termite preferences. Termite foraging increases as they start to colonize CWD [45], followed by small oscillations. The increase in the population of termites was estimated as:

$$NT = A \times (G - D) \times f_{tm} \qquad (6)$$

where *NT* is the net increase in number of termites; *A* is the maximum population (individuals) [46–49]; *G* is the birth rate [50, 51]; *D* is the death rate; $f_{tm}$ is the coefficient of temperature (5–40°C) [52] and moisture (10–99%) [53, 54]. Accordingly, the termite population will reach maximum when birth and death are equal, and decline when the mortality is higher than the birth rate. We assumed that termites will leave when the water table level reaches the soil surface (CWD decomposition in long-term inundation areas was not considered although termites can survive on trees in wetlands, such as mangroves), and that the birth rate was constant in a specific environment [55, 56]. We also assumed that the death and/or leaving rate increased with time in a specific colony location when the food or living conditions for termites were limited. Accordingly, termites on CWD (*B*) was:

$$B = \sum NT_i \qquad (7)$$

where $NT_i$ is the net increase in termite population in time *i*, $NT_i \in (-A, A)$. If $NT_i$ were approximate to -A, all termites should have either died or moved, and *B* will be equal to 0 if $B \leq 0$, i.e., no termites.

Wood decomposing beetles were considered as a group. Although many beetles cannot directly consume CWD, they can play a role in fragmentation, helping reduce CWD mass, and then those fragments can be decomposed by fungi and bacteria. Beetles' role in CWD decomposition [39] was described by Eq (8):

$$MBeetle = B \times C \qquad (8)$$

where *MBeetle* is the CWD mass loss due to beetles (mg day$^{-1}$); *B* is the total biomass of beetles colonizing the CWD; *C* is the daily consumption coefficient of the beetles (mg day$^{-1}$). The biomass was estimated using Eq (9).

$$B = S \times f_{tm} \times y \qquad (9)$$

where *S* is the coefficient of available CWD surface area; $f_{tm}$ is the coefficient of temperature and moisture, the range of temperature is between 5 and 40°C with an optimal temperature of 25°C; *y* is a non-linear time function, $y \in [0, 1]$, reduced from 1 to 0 with an increase in time, year.

Although bacteria can decompose some components of CWD [57, 58] and have been reported to effect wood decomposition [2, 16, 23, 58–65], the role of bacteria in wood decomposition is unclear. Bacteria can occur in various stages of CWD decomposition and their richness and diversity increase linearly with decreasing the wood density during the decomposition [16], but changes in bacterial community richness and diversity were not related to variations in the fungal community [16]. Overall, the contribution of bacteria to CWD decomposition is thought to be small [60].

The CWD mass loss attributable to bacterial decomposition was described using Eq (10):

$$MBacteria = B \times C \qquad (10)$$

where *MBacteria* is the CWD mass loss due to bacterial decomposition (mg day$^{-1}$); *B* is the

total bacterial biomass (mg) on the CWD; and $C$ is the bacterial respiration coefficient. The bacterial biomass $B$ was calculated as,

$$B = m \times f_{tm} \times y \tag{11}$$

where $m$ is the coefficient of available C (DOC, dissolved organic C) and nitrogen (DON, dissolved organic nitrogen) for bacterial growth; $f_{tm}$ is the coefficient of temperature and moisture; $y$ is a nonlinear time function, $0 \leq y \leq 1$, increased from 0 to 1 with an increase in time, year. The coefficient $m$ is also related to the competition between bacteria and other decomposers.

### Incorporation of wood carbon into the soil

**Fragment decomposition.** Fragmented CWD (see Fragmentation) falls onto the forest floor where fine wood particles and their decomposition products (i.e., POC, particulate organic C) may be incorporated into the surface mineral soil. Since these fragments are mainly decomposed by bacteria and fungi, their decomposition is similar to fungi and bacteria as mentioned above (see Eqs 3, 4, 10 and 11).

**Dissolved organic carbon.** DOC generated from CWD is considered to be the main pathway of wood C incorporation into soils [66–68]. DOC is leached into the soil where a portion of the wood C is incorporated into soil C pools [69], and can be divided into two parts: C produced within the wood (CWDi), and C from the decomposition of log fragments (CWDl). This distinction is made because of differences in temperature and moisture within the substrates and the corresponding effects on the decomposition process for each.

The DOC produced from CWDi and CWDl decomposition was calculated, respectively, using Eq (12):

$$DOC_i = C_i \times M_i \times f_{tmi} \tag{12}$$

where $M_i$ is the consumed mass of CWDi or CWDl by decomposer $i$; $C_i$ is the coefficient for the decomposer $i$ generating DOC; $f_{tmi}$ is the coefficient of temperature and moisture. Because temperature and moisture influence DOC production [70], the $f_{tmi}$ in Eq 12 for fragment (CWDl) decomposition was also regulated by soil temperature and moisture rather than only by the air temperature and moisture. The total DOC resulting from CWD decomposition is

$$T_{DOC} = DOC_{log} + DOC_{frag} \tag{13}$$

where $T_{DOC}$ is the total DOC from CWD decomposition; $DOC_{log}$ and $DOC_{frag}$ were produced by CWDi and CWDl, respectively.

**Leaching.** Leaching is a physical process [71] that transports DOC from CWD into the soil. However, only a portion of the leached C from wood decomposition is incorporated into soil, as other portions of the DOC are oxidized in the soil or leached into subsurface water. DOC leached into the soil from CWDi and CWDl decomposition was estimated as:

$$L_{DOCi} = S \times R_i \times DOC_i \times f_{mi} \tag{14}$$

where $L_{DOCi}$ is the amount of DOC leached into soils by water (g) at time $i$; $S$ is the coefficient of effective surface of the CWDi, $S \in (0, 1)$; $DOC_i$ is the available DOC at time $i$; $f_{mi}$ is a nonlinear leaching coefficient, changing from 1 to 0 with time, i.e., $f_{mi} \in [0, 1]$, as part of the water can be retained in CWD at the beginning of precipitation; $R_i$ is the effective leachate (ml) and changes with time, i.e.,

$$R_i = P_i \times f_i \tag{15}$$

where $R_i$ is the available water at time $i$ (hour); $P_i$ is the precipitation received by standing CWD or the throughfall received by downed CWD at time $i$ (cm), $P_i \in [0, 0.5]$; $f_i$ is a non-linear coefficient, changing from 1 to 0 with time, i.e., $f_i \in [1, 0]$; $i = 1, 2, \ldots, H$ that are the hours of leaching, and the $H$ was computed by the following equation,

$$H = P \div r \tag{16}$$

where is $H$ is the length of time (hours); $P$ is the daily precipitation or throughfall (cm); $r$ is the precipitation rate (cm hr$^{-1}$), the maximum $r$ is 0.5 cm hr$^{-1}$.

The DOC leaching through the soil profile and lost to the water table was calculated by Eq 17,

$$S_{iDOC} = DOC_i \times W_i \times l_i \tag{17}$$

where $S_{iDOC}$ is the DOC loss from soil layer $i$ to next layer (2 cm intervals) or aquifer when the soil layer is approximate to the level of the water table; however, if water table is lower than 50 cm below the surface, DOC generated from CWD decomposition at soil layers $\geq 50$ cm in depth was considered as a loss; $DOC_i$ is the concentration of DOC at the layer $i$; $W_i$ is the moveable water at the layer; $l_i$ the leaching coefficient. The time step for leaching is hourly.

The DOC incorporated into soils is the difference between the amount produced and leached from logs and the amount of loss from soils by leaching (Eq 17). The subsequent fate of the DOC retained in the mineral soil is not considered further in this model.

## Sensitivity analysis

CWDDAT was parameterized to analyze the sensitivity of the CWD decomposition processes to the ecological drivers used as model inputs: 1) CWD properties, including the size, species, mass, and position (standing vs downed), and 2) ambient conditions, including climate, soil properties, land cover (vegetation), and geographical location and geomorphic altitude. Climate data include daily minimum and maximum temperature, daily precipitation (rainfall and snowfall), evapotranspiration, and daily PAR (Photosynthetically active radiation). Land cover and soil characteristics from the Santee Experimental Forests in South Carolina in USA (green square in Fig 1) were used as the basis for this analysis because it has well documented measures of vegetation, soils, hydrology and climate over 80 years [72].

Climatic data obtained from the Daymet database [35] for eighty-nine points in North America (Fig 1) were used to analyze model sensitivity to temperature, precipitation, snowfall, latitude and altitude. The climatic datasets from Daymet cover a large spatial scale (14° to 65° N latitude and 58 to 139° W longitude), a long time period (1980–2017), and altitude ranging from 3 m to 2,824 m above mean sea level, mean annual temperature from -11.8 to 26.5°C, and mean annual precipitation from 181 to 6,143 mm.

Assumptions used for the sensitivity analysis included: (1) all sites were upland forests; (2) the forests were mature; (3) soils for all sites were loam; (4) CWD species group was either hardwood or softwood to assess the differences in CWD decomposition among the tree species groups (Table 1); (5) CWD was either standing (snag) or lying on the soil surface (downed) to determine the differences in decomposition between positions; and (6) CWD was a mixture of hardwoods (50%) and softwoods (50%) to analyze the sensitivity of CWD decomposition to climate because tree species are mixed in many forests. We assumed that the CWD was initially sound, or in decomposition class 1. Since the size of CWD has been defined as >2.5 cm or $\geq 7.5$ cm in diameter [2], we set >4.0 cm in diameter to classify woody residue as CWD, and used six size classes in the sensitivity analysis: 4.0–7.5, 7.5–15.0, 15.0–22.5, 22.5–30.0, 30.0–37.5, and >37.5 cm.

**Table 1. Wood mass (kg) used to analyze the model sensitivity to climate[*].**

| Position | Species group | Size 1 | Size 2 | Size 3 | Size 4 | Size 5 | Size 6 |
|---|---|---|---|---|---|---|---|
| Downed | Soft | 2,500 | 5,000 | 12,000 | 36,000 | 12,000 | 7,500 |
| | Hard | 2,500 | 5,000 | 12,000 | 36,000 | 12,000 | 7,500 |
| Standing | Soft | 2,500 | 5,000 | 12,000 | 36,000 | 12,000 | 7,500 |
| | Hard | 2,500 | 5,000 | 12,000 | 36,000 | 12,000 | 7,500 |

[*]size 1: 4.5–7.49; size 2: 7.5–14.99, size 3: 15.0–22.49, size 4: 22.5–29.99, size 5: 30.0–37.49, and size 6: ≥37.0 cm in diameter.

The effects of each of the decomposer groups and their combinations (synergies) were assessed by excluding those not being considered to affect the decomposition processes. However, physical and chemical processes, such as fragmentation and leaching, were included when assessing synergistic effects, because these processes exist irrespective of the decomposer community. The climate data at Santee (green square in Fig 1) were used to analyze the sensitivity of CWD decomposition to each of wood-decomposer type individually and in combination.

CWD with different size classes and different positions (downed and standing) was used to analyze the model sensitivity to these factors (Table 1). To analyze the sensitivity to climate, the CWD was also mixed with different size classes from size 1 to size 6 at the ratios of 3.3, 6.7, 16.0, 48.0, 16.0 and 10.0% of the total mass, respectively, with a mean of wood diameter of 26 cm. Eight wood density classes were used to analyze the effect of wood density on decomposition, ranging from 0.3 g cm$^{-3}$ to 1.0 g cm$^{-3}$ with an interval of 0.1 g cm$^{-3}$. The simulation decomposition period was 100 years, beginning from the year of tree mortality. This time span was selected to give enough time for the wood decomposition process to be completed in a cold climate, such as in boreal forests. The time step of this model was hourly for biological respiration and leaching, and daily for other processes.

## Statistical analysis

Univariate and multivariate linear and non-linear regressions were used to analyze the sensitivity of CWD decomposition to the main ecological drivers used as model inputs. Student's t-test was used to determine whether or not there were differences in CWD decomposition constants obtained from different decomposition models suggested by some studies [26] and used in this study (see the Eqs 20–24 below). To increase the reliability of results from model sensitivity analysis, the level of statistical significance was set to α = 0.02 to determine whether the rates of CWD decomposition were sensitive to one or multiple eco-environmental factors, rather than α = 0.05; accordingly, the use of "significant" or "significantly" indicates $P \leq 0.02$ hereafter.

The time span to CWD mass loss of 50% (half time, $T_{50}$) and years to the mass loss of 95% ($T_{95}$) were used to describe the differences in CWD decomposition among the ecological drivers, and calculated, respectively, as

$$T_{50} = 0.69315 \div k \tag{18}$$

$$T_{95} = 2.99573 \div k \tag{19}$$

where $k$ is the decomposition constant (y$^{-1}$).

The decomposition constant $k$ of CWD was calculated using different equations because the CWD decomposition may not follow a perfectly exponential model [7, 26, 29, 73].

Accordingly, four sets of CWD decomposition constants were calculated to find an optimal decomposition constant. We first assumed that the CWD decomposition followed a single exponential model as in Eq 20,

$$M_t = M_0 \times e^{-k_1 \times t} \tag{20}$$

where $M_t$ is the mass remaining at time $t$ (years); $M_0$ is the initial mass. Accordingly, the decomposition constant $k_1$ can be calculated for each site using following equation, i.e.,

$$k_1 = \frac{1}{n} \sum_1^n \{ [\ln\ (M_0) - \ln\ (M_t)] \div t \} \tag{21}$$

where $ln(M_0)$ and $ln(M_t)$ are natural logarithmic values of the initial CWD mass ($M_0$) and the remaining mass ($M_t$) at time $t$ (years); $t = 1, 2, 3, \ldots, n$ that is the simulation time span, years. The $k_1$ used for a specific site was averaged from all simulated years at the site.

Regressions are widely used to fit observations for empirical equations. Accordingly, we employed regressions to obtain the constants of CWD decomposition using data from simulations for the sensitivity analysis. The decomposition constants $k_2$ and $k_3$ were obtained by fitting the mass remaining from CWD decomposition simulations for each site using single exponential functions; the $k_3$ was obtained from fitting with forcing the intercept of the exponential function to $M_0$ (equal to the initial mass); but $k_2$ was without forcing the intercept such that it may be smaller or greater than the initial CWD mass. Accordingly, the fitted single exponential equations had the form

$$\ln\ (M_t) = f - k_2 \times t \tag{22}$$

where $M_t$ is the remaining mass (known); $f$ is a regression coefficient, and exp($f$) is equal to the intercept of the fitted exponential function; $t$ is time (known, years); accordingly, linear regression can be used to obtain the coefficients $f$ and $k_2$ when $Y_t = \ln(M_t)$. If the intercept of the exponential function is forced to the initial CWD mass, i.e., $f$ in Eq 22 equals $\ln(M_0)$, thus,

$$\ln\ (M_t) = \ln\ (M_0) - k_3 \times t \tag{23}$$

This equation (Eq 23) appears similar to Eq 21, but the constant $k_3$ is not averaged from annual decomposition, it is obtained using the linear regression, like the method to obtain $k_2$ in Eq 22.

The constants of CWD decomposition were also calculated using a combination of power and exponential functions (Eq 24) to assess CWD decomposition that did not follow a single exponential function, i.e.,

$$\ln\ (M_t) = \ln\ (M_0) + k_5 \times \ln\ (t + 1) - k_4 \times t \tag{24}$$

where $M_t$, $M_0$ and $t$ are as same as in Eq 20; and $k_4$ and $k_5$ are coefficients determined by a multivariate regression. Because of the multiple $k$s, $T_{50}$ and $T_{95}$ were estimated using iteration for the decomposition model as Eq 24.

## Results

Results from sensitivity analysis to model inputs, which were wood properties and ambient conditions, including temperature, precipitation, snowfall, and geographic and geomorphic information at eighty-nine sites, are presented below. Results for $k$ values calculated using different equations and the corresponding half-life of CWD decomposition are in S1 Fig and S3 Table.

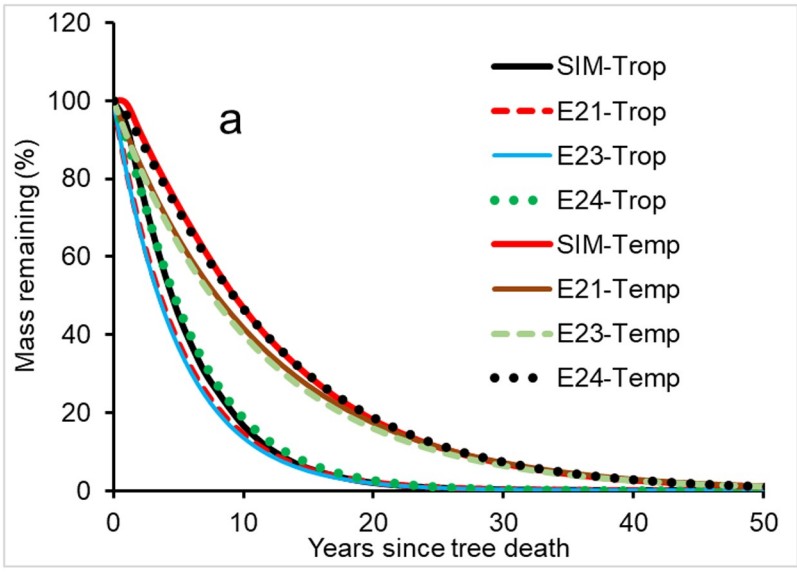

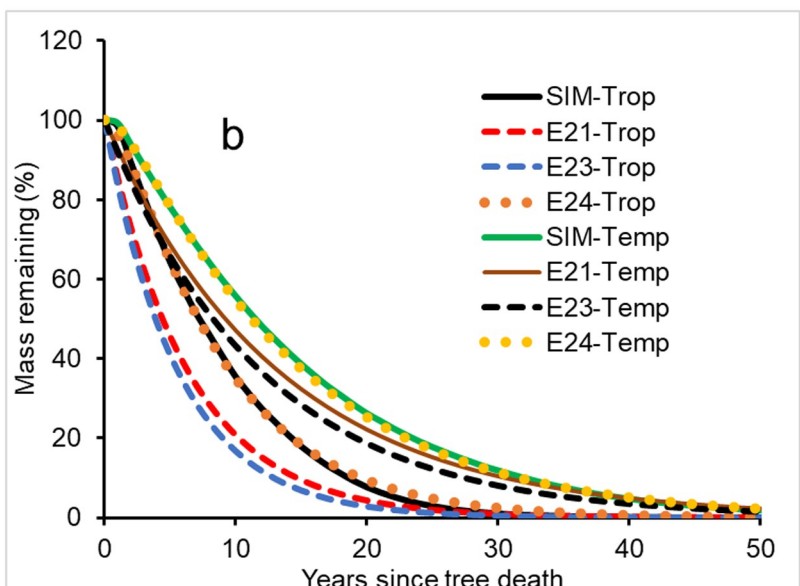

**Fig 3. Differences in CWD mass loss over time with the decomposition of downed deadwood (a) and standing deadwood (b) under tropical (S84, 14.2° N) and cold-temperate (S49, 49° N) climatic conditions.** SIM, based on the result from the sensitivity analysis and E21 –E24 from the mass remaining calculated using the coefficients from the Eqs 21–24. Intercepts obtained from Eq 22 were not forced to the initial mass such that they can be smaller or larger than 100% of the initial mass. -Trop, tropical; -Temp, temperate; the data points for each dataset were 100, i.e., $n = 100$.

## Disparities among the decomposition constants obtained from different decomposition models

There are differences in *k* values (S3 Table) obtained from different calculation equations (see Eqs 20–24). The decomposition constants calculated using different equations showed small differences in downed wood decomposition at specific sites (Fig 3a), but large differences were found in standing CWD decomposition (Fig 3b). Because Eq 24 was developed to match the

simulated CWD mass remaining, the calculated mass attenuation with time using the decomposition constants from Eq 24 for both downed and standing CWD was approximate to the CWD mass attenuation trend for sensitivity analysis.

The time to decompose 50% of the CWD ($T_{50}$) was calculated using different $k$-based models for assessing the sensitivity of CWD decomposition to climate and showed considerable differences among the eighty-nine sites (S1 Fig). The smallest CWD decomposition constants occurred at the site S30 (65.0°N and 105°W) where the $T_{50}$ values were the largest among the eighty-nine sites. The largest decomposition constants (smallest $T_{50}$) occurred at site S87 (15.8°N and 92.8°W; S1 Table and S1 Fig), regardless of which equation was used to obtain the constant (S3 Table).

There were differences in $T_{50}$ among the equations of $k$-based calculations (Eqs 21–24) ($P > 0.02$), based on Student's t-test ($58.38 > |t| \geq 2.96$, $|t| > t_{critical} = 2.37$). Accordingly, the constants of CWD decomposition mentioned and used below would be mainly based on Eq 24 because this equation was developed based on data from this sensitivity analysis.

The $T_{50}$ among the eighty-nine sites varied widely, ranging from 4.4 to 67.3 years with a mean of 12.6±9.8 (mean ± SD) years for the downed CWD decomposition, using the constants calculated from Eq 24 (coefficient $k_4$ and $k_5$), and ranging from 6.8 to 65.2 years with a mean of 15.1±9.1 years for standing CWD using the same decomposition model. These metrics demonstrate substantial differences in CWD decomposition among the sites because of the different climatic conditions.

## Sensitivity of CWD decomposition to climate

Results from sensitivity analyses showed that climate was the most important variable affecting CWD decomposition. The trends in wood mass loss for downed CWD decomposition at five sites with differing climatic zones (tropical, subtropical, temperate and boreal) are presented in Fig 4a, and for standing CWD decomposition in Fig 4b. In both the rates of wood mass loss in temperate and boreal zones were substantially smaller than the rates in tropical and subtropical zones ($P < 0.02$), but the difference was statistically small or insignificant among some tropical sites and/or subtropical locations ($P > 0.05$). The differences among the climatic zones were regulated by multiple factors, including temperature, rainfall, and snowfall.

**Temperature.** The CWD decomposition constants ($k_4$ and $k_5$) increased non-linearly with increases in temperature ($n = 89$, $R^2 > 0.76$ for downed CWD; $R^2 > 0.66$ for standing CWD, $P < 0.001$; Fig 5), i.e.,

$$k_4 = C \times e^{D \times T} \tag{25}$$

where $k_4$ is the exponent of the exponential function in Eq 24, $C$ and $D$ are coefficients, $C = 0.0575$ and $0.0502$ for downed and standing CWD, respectively, $D = 0.0497$ and $0.0402$, respectively; $T$ is annual mean temperature (°C). Similar to $k_4$ values calculated using Eq 24, $k_5$ values for both downed and standing CWD also increased exponentially with temperature ($n = 89$, $R^2 = 0.61$ and $0.53$ for downed and standing CWD, respectively, $P < 0.001$),

$$k_5 = f \times e^{g \times T} \tag{26}$$

where $f$ and $g$ are coefficients, $f = 0.0616$ and $0.0651$ for downed and standing CWD, respectively, $g = 0.033$ and $0.034$, respectively. However, constant $k_4$ of Eq 24 for standing CWD decomposition ($0.078±0.0412$) was smaller than that for downed CWD ($0.1067±0.0616$), while constant $k_5$ for standing CWD ($0.0912±0.042$) was similar to that for downed ($0.082±0.041$). The $k_4$ and $k_5$ for both positions of CWD varied little with temperature when annual mean air

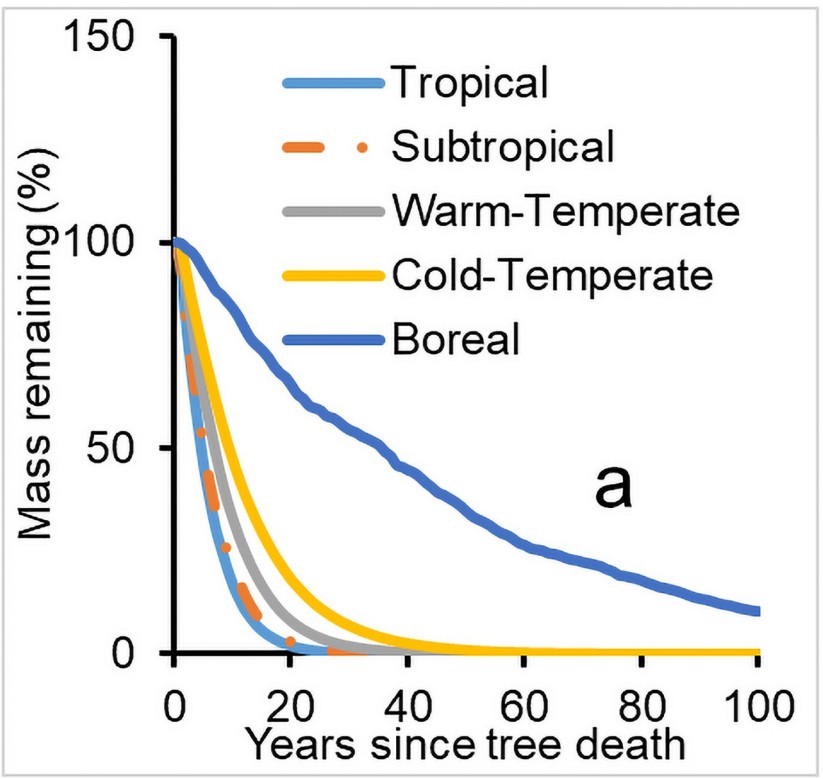

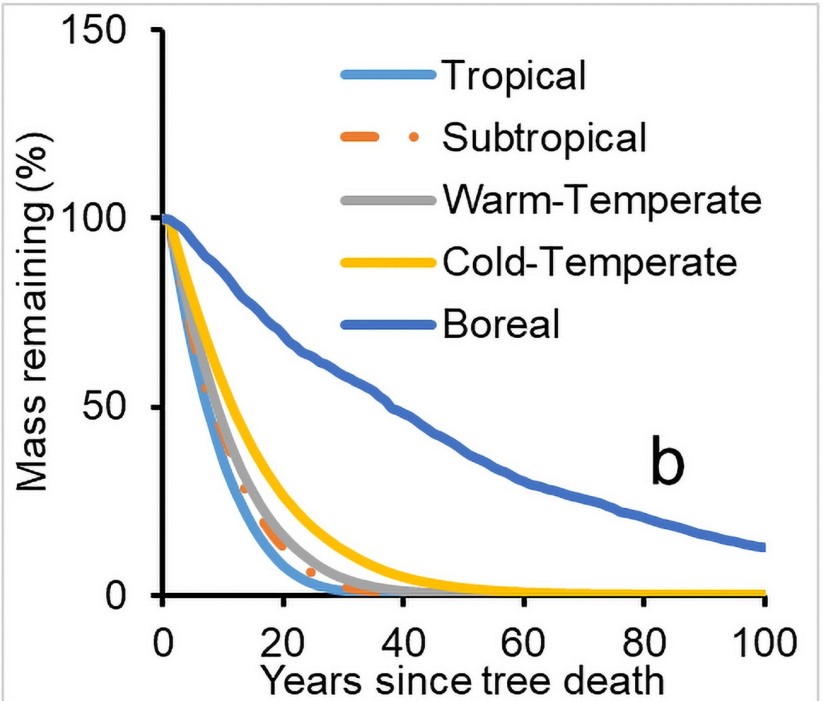

**Fig 4. Decomposition of downed (a) and standing (b) CWD under different climatic zones.** The latitudes for these five sites are 14.2, 29.0, 43.0, 49.0 and 65.0˚N, respectively.

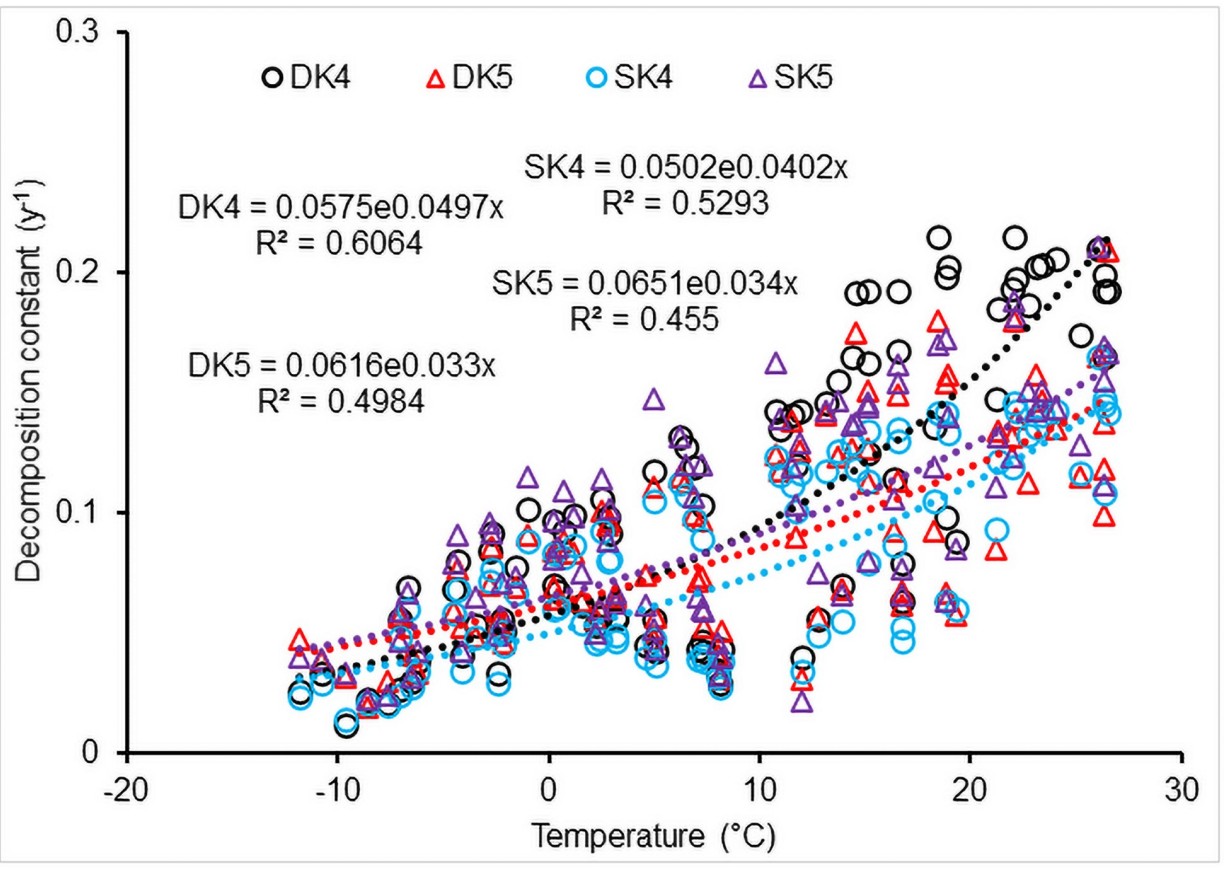

**Fig 5. Impact of temperature on CWD decomposition.** DK4 and DK5 calculated on the basis of Eq 24 for downed deadwood; SK4 and SK5 for standing deadwood.

temperature was $< -7°C$ or $> 25°C$, although the $k_4$ and $k_5$ values significantly increased exponentially with temperature in general (Fig 5).

DOC from downed CWD decomposition leaching into soil increased exponentially as temperature increased ($R^2 = 0.71$, $n = 89$, $P<0.001$), and the loss from soils due to leaching exhibited an exponential response (Fig 6). The relationship between the net DOC incorporation into soils from downed CWD decomposition and annual air temperature was

$$DOC_{ind} = a \times e^{b \times T} \tag{27}$$

where $DOC_{ind}$ is the rate of total DOC from downed CWD decomposition incorporated into soils to the initial mass of the downed CWD (g kg$^{-1}$); $T$ is annual mean temperature (°C); $a$ and $b$ are constants, $0 \leq b < 1.0$, and $a$ is related to wood species and precipitation. Although DOC production from standing CWD decomposition was not less than that from downed CWD (Fig 7), the incorporation of DOC from standing CWD was a cubic polynomial increase with temperature ($P<0.02$), i.e.,

$$DOC_{ins} = a_1 \times T^3 - a_2 \times T^2 + a_3 \times T + C \tag{28}$$

where $a_i$ and $C$ are coefficients related to precipitation and wood species group of CWD; $DOC_{ins}$ is the rate of DOC incorporated into soils to the initial mass of the standing CWD (g kg$^{-1}$). The incorporated DOC from standing CWD was lower than the DOC from downed

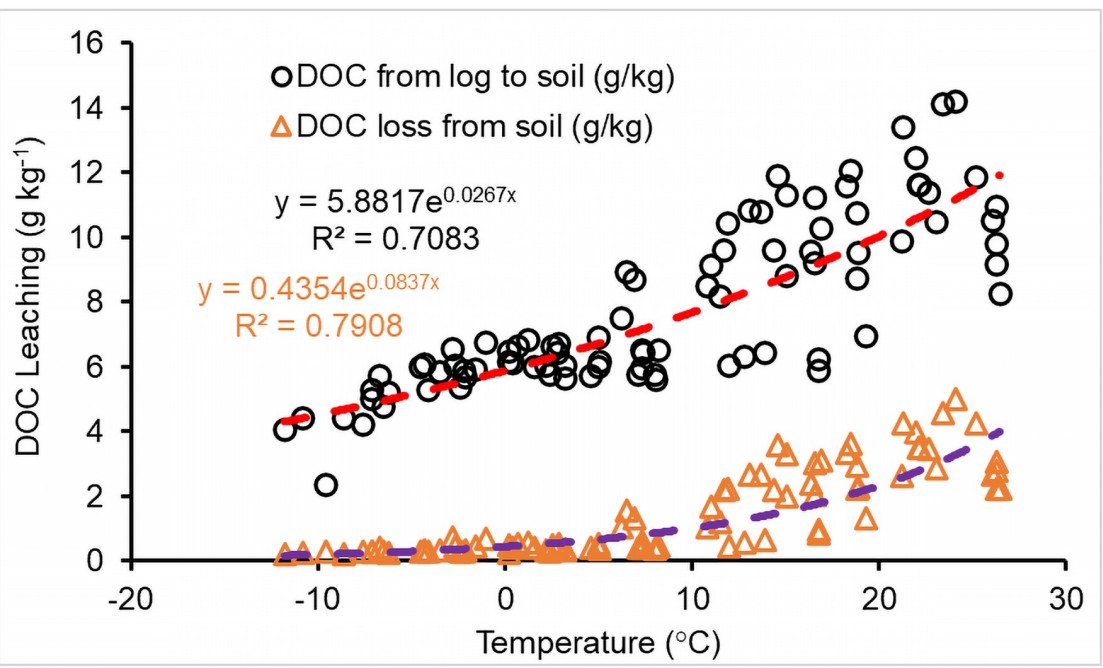

**Fig 6. Impact of temperature on dissolved organic carbon (g C mass per kg CWD) leaching during downed CWD decomposition.** From-log-to-soil indicates total DOC leached from downed CWD. Loss-from-soil indicates the portion of DOC from log that was lost to outflow or groundwater due to soil leaching.

CWD. Less DOC went to the soil from standing CWD because most of the DOC was decomposed or oxidized before leaching, and the C remaining in soils would then be gradually lost to subsequent soil leaching or microbial activity in the soil.

POC (particulate organic C ≤ 2mm in diameter) can be incorporated into soils by leaching, but it can be gradually decomposed after it gets into soils. The total (accumulative) POC from CWD decomposition within the simulation period decreased non-linearly with an increase in temperature (Fig 8). POC from standing logs was more than that from downed CWD, which

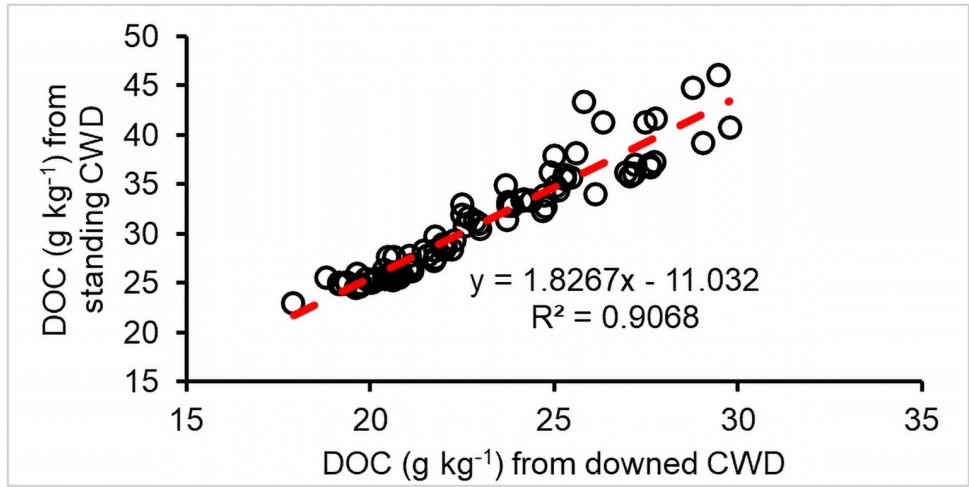

**Fig 7. DOC generated from downed CWD versus standing CWD (g C per kilogram CWD C).**

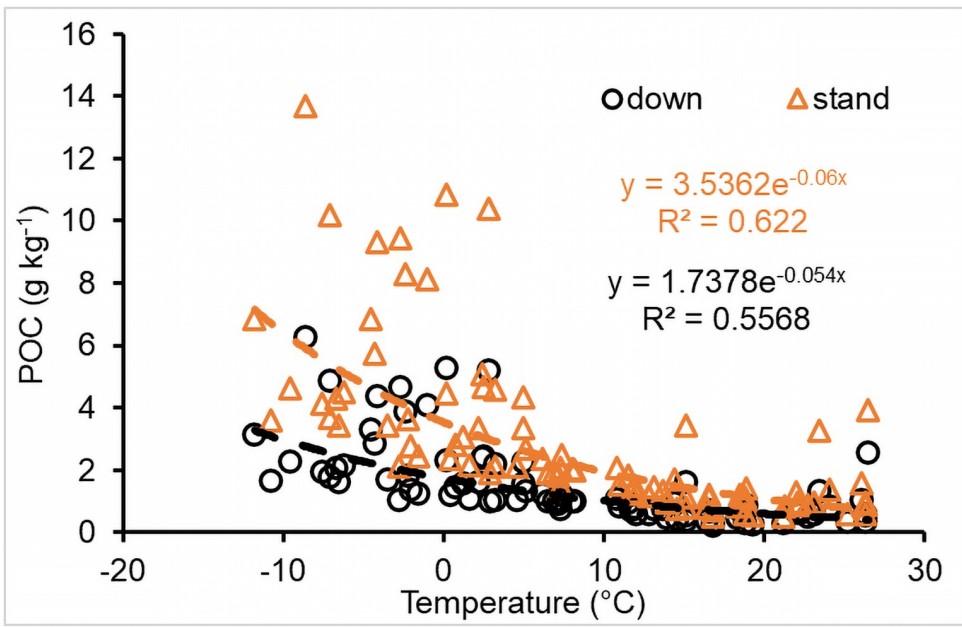

**Fig 8. Impact of temperature on POC (particulate organic carbon, g C mass per kg CWD) incorporated into soils.**

was related to microsite variability, such as diurnal temperature differences, but these wood fragments would be gradually decomposed.

**Precipitation.** The decomposition constants ($k$) of both standing and downed CWD increased significantly with logarithmical increments in annual precipitation (Fig 9; $n = 89$; $R^2 = 0.56$ and $0.64$ for downed and standing CWD, respectively, $P<0.001$), following Eq 29,

$$k_{4i} = C_i \times ln(Pcp) + D_i \tag{29}$$

where $C_i$ and $D_i$ are coefficients corresponding to the $k_{4i}$ ($i = 1$ for downed CWD and $i = 2$ for standing); $Pcp$ is annual precipitation (mm). Similar to the effect of temperature on $k_4$ values of Eq 24, the exponent $k_5$ of the Eq 24 also increased significantly with a logarithmic increase in precipitation for both downed and standing CWD decomposition (n = 89, $R^2 = 0.58$ and $0.59$ for downed and standing CWD, respectively, $P<0.001$). This relationship between precipitation and the CWD decomposition constant indicated that CWD decomposition was highly sensitive to precipitation, especially at values less than 2000 mm y$^{-1}$ (Fig 9).

The correlation between precipitation and DOC incorporated into soils for both downed and standing CWD (Fig 10) increased non-linearly (in power function) with precipitation ($P<0.001$), indicating that precipitation is an important factor impacting DOC incorporation into soils during CWD decomposition.

**Snowfall.** CWD decomposition was significantly correlated to annual snowfall ($P<0.001$). The CWD decomposition decreased significantly with an increase in logarithmic annual snowfall (Fig 11; n = 69, sites with no snow excluded). The sensitivity analysis indicated that annual snowfall largely affected CWD decomposition when annual snowfall less than 200 kg m$^{-2}$, but the decomposition was less sensitive to annual snowfall over 200 kg m$^{-2}$. The low sensitivity to snowfall over 200 kg m$^{-2}$ was related to snow's insulating effect.

The effect of snow on DOC incorporation into soils differed between downed and standing CWD (Fig 12). The incorporation of DOC generated from downed CWD decomposition significantly decreased with an increased in logarithmic snowfall ($P<0.001$), which was

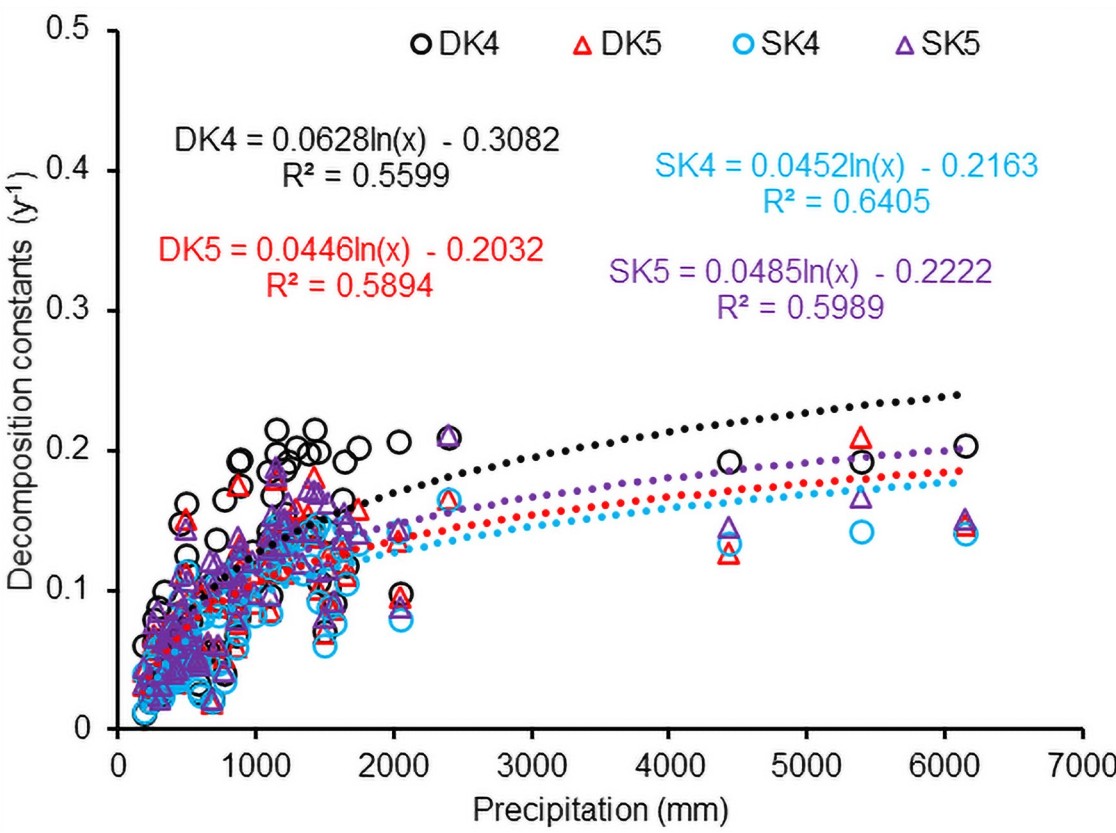

**Fig 9. Sensitivity of CWD decomposition to mean annual precipitation (mm).** DK4 and DK5 calculated on the basis of Eq 24 for downed CWD; SK4 and SK5 for standing CWD.

consistent with the trend in the decomposition rates, and there was no a substantial difference for annual snowfall over 200 kg m$^{-2}$. However, the incorporation of DOC released from standing CWD decomposition increased linearly with increasing annual snowfall, but the increment was small, adding only about 2 mg of DOC for each additional kg m$^{-2}$ of snowfall (Fig 12).

The incorporation of POC into soils generated from both downed and standing CWD increased linearly with increasing annual snowfall ($n = 69$, $R^2 = 0.69$ and $0.66$ for downed and standing CWD), i.e.,

$$POC = a \times Snow + b \tag{30}$$

where $a$ and $b$ are coefficients, and the slope $a$ was 7.6 and 15.7 mg C kg$^{-1}$ of snowfall for downed and standing CWD, respectively; *Snow* is annual snowfall (kg m$^{-2}$).

## Impact of latitude on CWD decomposition

The sensitivity analysis showed that both of the constants $k_4$ and $k_5$ of Eq 24 for downed CWD decomposition decreased with a logarithmic increase in latitude ($n = 89$, $R^2 > 0.63$, $P<0.001$). Correspondingly, the relationship of CWD decomposition constants to latitude can be described by Eq 31,

$$k_j = a \times \ln{(LAT)} + c \tag{31}$$

where $k_j$ is the constant, $k_4$ or $k_5$, in Eq 24 for downed CWD decomposition; *Lat* is the latitude

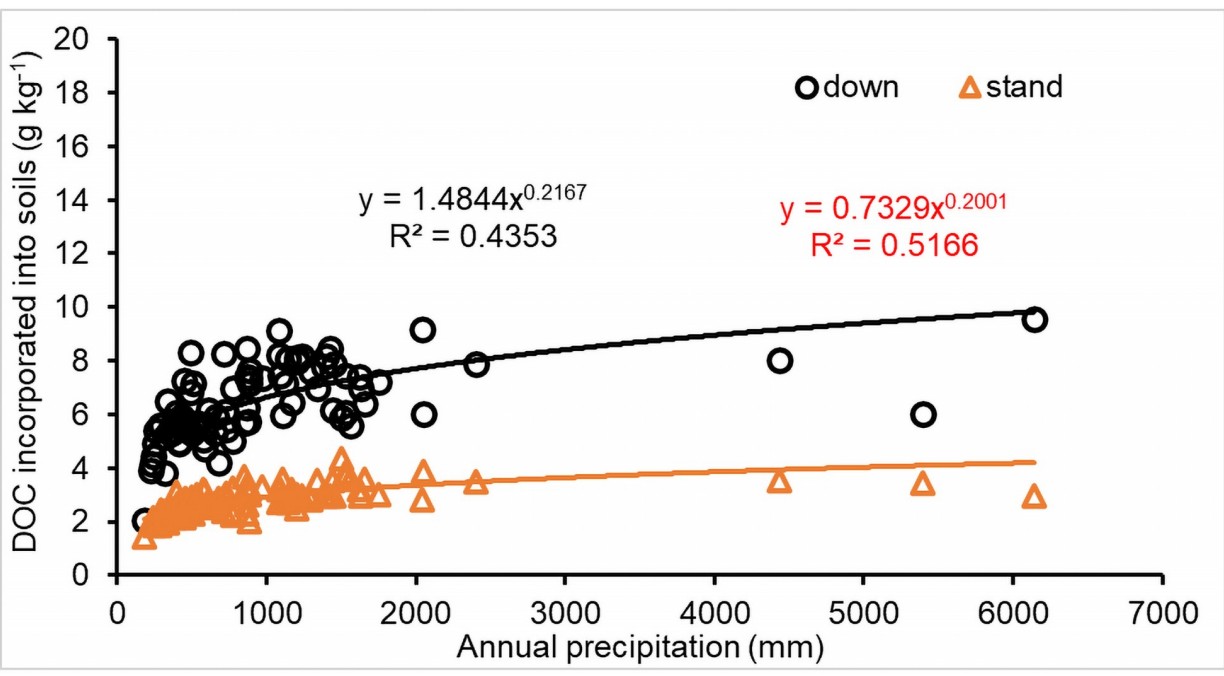

**Fig 10. Impact of precipitation (mm) on DOC incorporated into soils (g C mass per kilogram CWD) for downed (down) and standing (stand) CWD decomposition.**

of a location (degrees, absolute value); and $a$ and $c$ are constants, $a$ = -0.112 and -0.076 for $k_4$ and $k_5$, respectively. In contrast, both constants $k_4$ and $k_5$ of Eq 24 for standing CWD decreased linearly with increasing latitude ($P<0.001$), i.e.,

$$k_j = f \times Lat + g \tag{32}$$

where $k_j$ is the constant, $k_4$ or $k_5$, in Eq 24; and $f$ and $g$ are coefficients, $f$ = -0.002 and -0.0021 for $k_4$ and $k_5$, respectively. These metrics indicate that CWD decomposition was sensitive to geographic latitude, but there was a difference in the sensitivity between downed and standing CWD, with downed CWD logarithmically responding to the changes in latitude, and standing CWD responding linearly.

DOC and POC incorporated into soils also responded to variation of geographic latitude. The incorporation of DOC from downed CWD decomposition decreased exponentially ($P<0.001$) with latitude increase (Fig 13). However, DOC from standing CWD had a cubic polynomial relationship with latitude (Fig 13), reflecting a difference in DOC incorporation into soils between DOC sources that were generated from downed or standing CWD.

POC from both downed and standing CWD decomposition responded exponentially to an increase in latitude ($n$ = 89, $R^2$ = 0.471 and 0.521 for the correlations of latitude to POC from downed and standing CWD decomposition, respectively), the relationship can be described by

$$POC = a \times e^{b \times Lat} \tag{33}$$

where $a$ and $b$ are coefficients, $a$ is 0.266 and 0.44 g kg$^{-1}$ for downed and standing CWD, respectively, and $b$ is 0.0374 and 0.0402, respectively; $Lat$ is number of degrees of the latitude (absolute value).

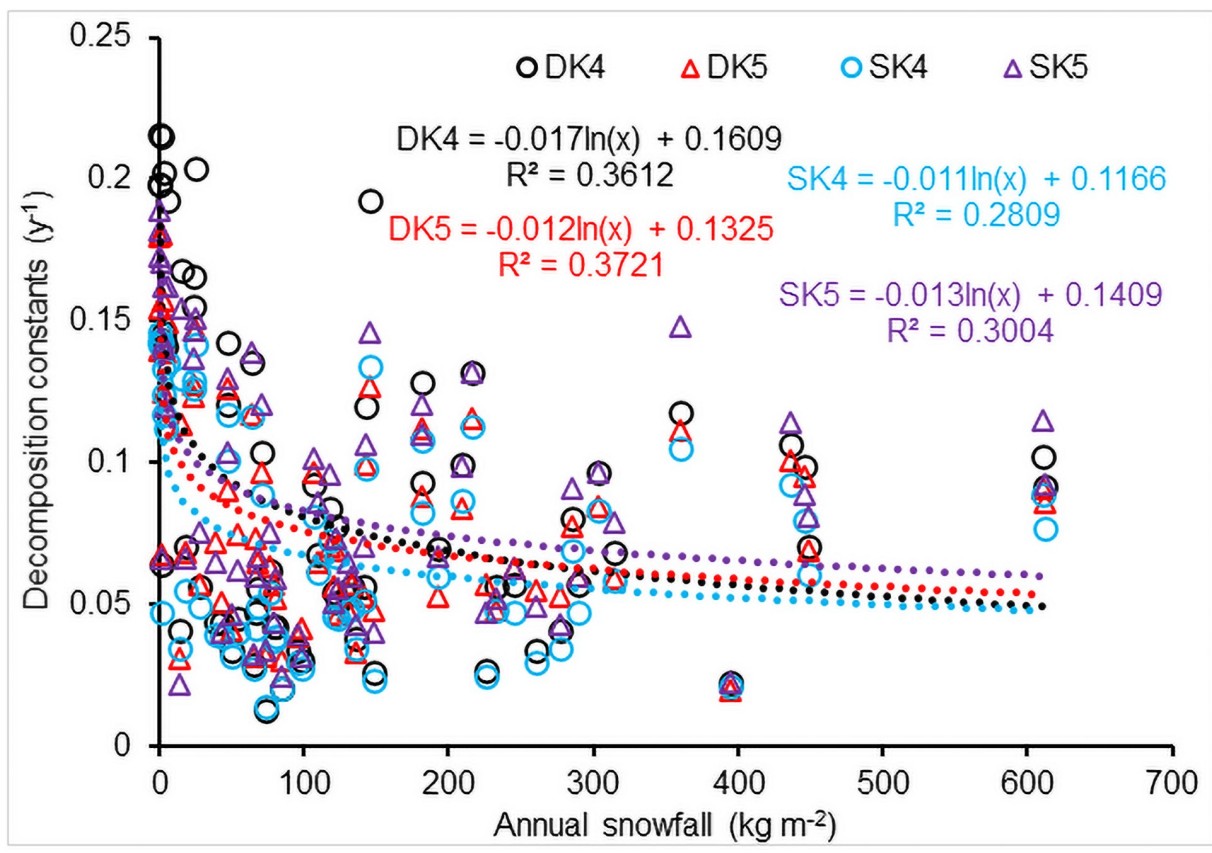

**Fig 11. Effect of snow on CWD decomposition ($n$ = 69, sites with no snow excluded).** DK4, DK5, SK4 and SK5, calculated using Eq 24 for downed and standing CWD, respectively.

## Effect of altitude on CWD decomposition

The results from the sensitivity analysis showed that CWD decomposition rate was correlated to geomorphic altitude. Fig 14 indicated that the decomposition constant decreased significantly ($P<0.001$) with an increase in logarithmic altitude (m), i.e.,

$$k_4 = a \times \ln{(ALT)} + c \qquad (34)$$

where $a$ and $c$ are coefficients, the slope $a$ was -0.017 and -0.012 for downed and standing CWD, respectively. Similar to the $k_4$ in Eq 24, the exponent $k_5$ significantly decreased with altitude ($P<0.001$), and the slope $a$ was -0.010 and -0.013 for downed and standing CWD, respectively. These metrics indicated that the CWD decomposition rate was correlated to geomorphic altitude, however, DOC and POC generated from downed and standing CWD decomposition was not correlated to geomorphic altitude ($P>0.02$).

## Contributions of decomposer groups

Assumed isolation methods (see Sensitivity Analysis) were used to assess the roles played by each wood-decomposer group and their synergisms in CWD decomposition. There were differences in CWD decomposition performed by each of the main decomposers and their synergies. Fig 15 presents the CWD mass attenuation with time based on the CWD decomposition performed by different decomposers and their synergistic behaviors at a subtropical site

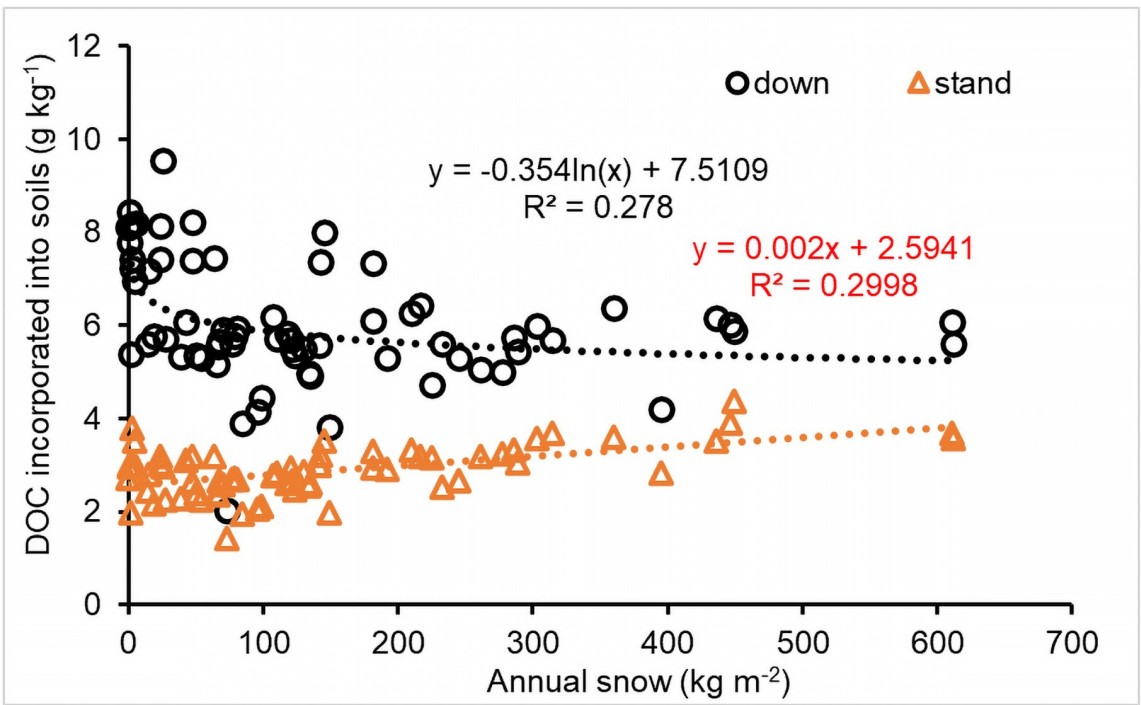

**Fig 12. Impact of snow on DOC incorporation into soils (g C mass per kilogram CWD) for snowfall >0.**

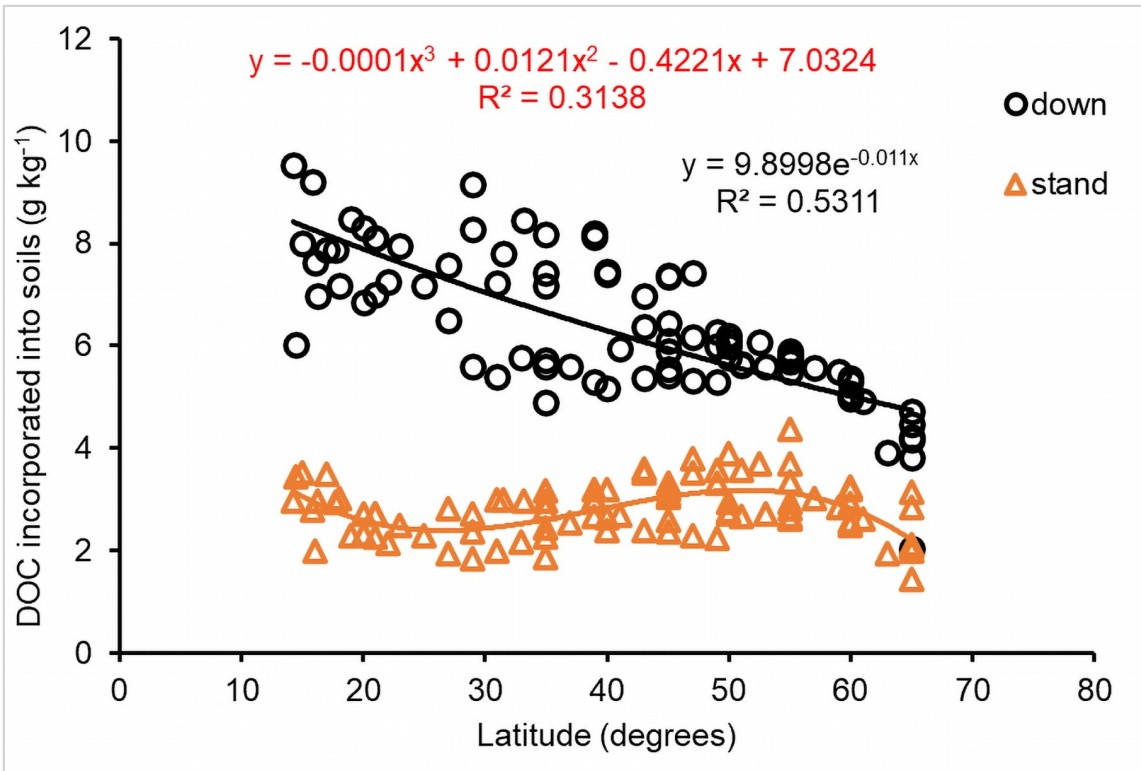

**Fig 13. Impact of latitude on DOC incorporated into soils (g C mass per kilogram CWD).**

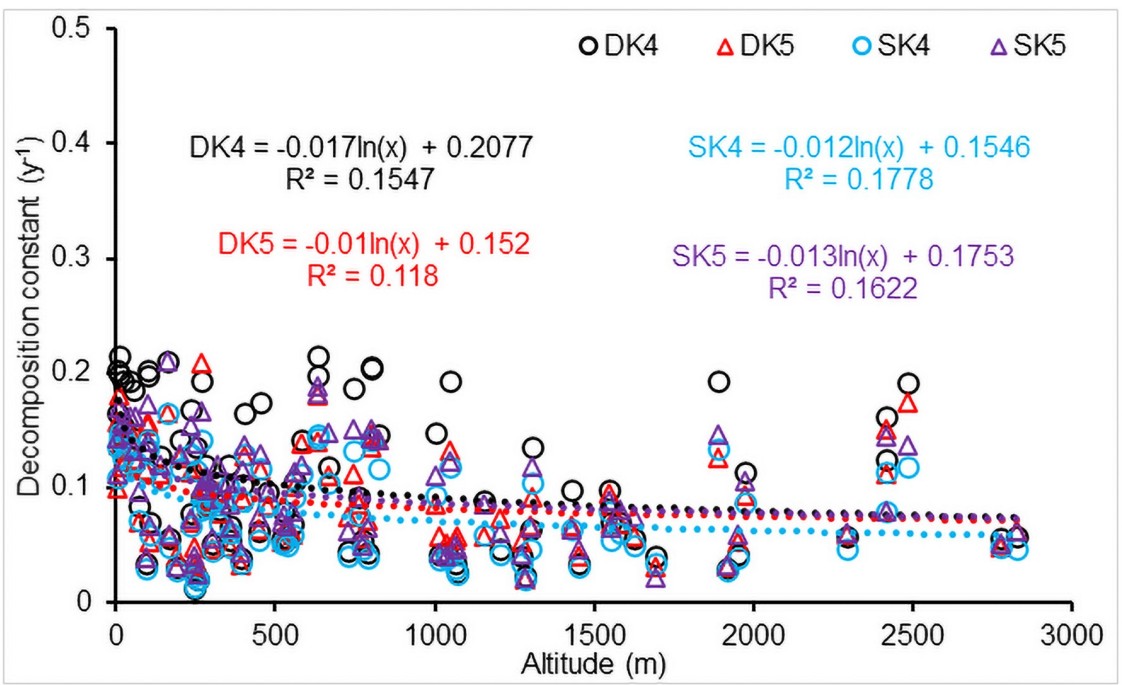

**Fig 14. Effect of altitude on CWD decomposition.** DK4, DK5, SK4 and SK5 calculated using Eq 24 for downed and standing CWD, respectively.

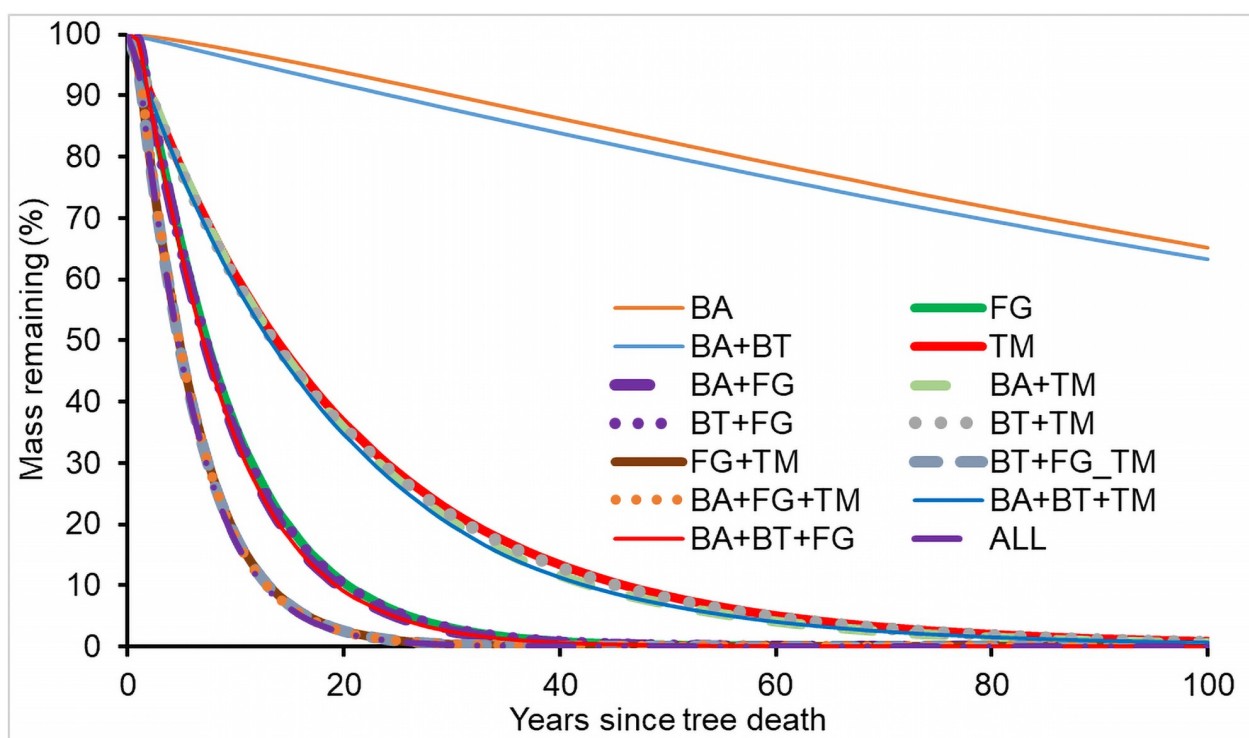

**Fig 15. Roles in downed CWD decomposition played by different decomposers and various synergies of different decomposers under the subtropical climatic conditions at Santee, South Carolina, USA.** Decomposers: BA, bacteria; BT, beetles; FG, fungi; TM, termites; ALL, BA+BT +FG+TM.

(Santee Experimental Forest, site S01, green square in Fig 1). The rate of CWD mass loss from the synergic effects of all decomposers combined (bacteria + beetles + fungi + termites) was the highest. There were only four sets of important synergies of various decomposers under the ecological conditions at Santee, and all of these four synergisms involved termites and fungi, indicating that these two decomposers were the most important factors governing wood decomposition at Santee, and other decomposers had a very small roles in CWD decomposition in this subtropical environment.

The rates of downed CWD mass loss to respiration of fungi and termites at Santee were 68.1 and 28.5% of the total biological respiration, respectively, only 3.4% of total respiration was contributed by beetles with bacterial synergy. However, the rates of standing CWD mass loss to the respiration of fungi and termites were 89.9 and 5.5% of the total, respectively, indicating a small effect of termites on standing dead trees. These metrics indicated that fungi played the most important role in CWD decomposition at Santee, although Santee is located in the area with the highest risk of termites in USA [74].

The respiration of different decomposers varied largely among the 89 sites (S2 Fig). The contribution of termites to the total biological respiration from decomposing downed CWD at the eighty-nine sites varied from 0.0% in boreal sites, > 25% at some subtropical areas, to 39.9% in tropical sites. The CWD decomposition rates contributed by fungi varied from 55.4 to 99.1% across the 89 sites, and the rates contributed by beetles with bacterial synergy were between 0.8 and 10.4% among the 89 sites. However, the respiration from standing CWD decomposition was different from downed CWD. Proportions of total biological respiration from standing CWD decomposition were 0.0 to 8.8% for termites, 83.9 to 98.8% for fungi, and 1.2 to 10.3% for beetles with bacterial synergy across the 89 sites.

## Effect of wood density

As expected, wood density influenced CWD decomposition (Column DS in Table 2). The rates of both downed and standing CWD decomposition significantly decreased with an increase in wood density (Fig 16). However, there was a difference in the sensitivity to wood density between downed and standing CWD. The decrement in the rates of downed CWD decomposition with wood density increase was less than that for standing CWD, indicated that standing CWD decomposition was slightly more sensitive to changes in wood density than downed CWD (Fig 16).

## Wood size affect

Constant $k_4$ from Eq 24 for CWD decomposition decreased with an increase in wood size, regardless of CWD position (standing or downed) or whether wood was softwood or hardwood (Column Size in Table 2; Fig 17a). The trend in the $k_4$ decreased generally with increasing wood size, following power functions for downed and standing CWD (Fig 17a). Constant $k_5$ from Eq 24 for CWD decomposition were slightly different from the $k_4$ from the same equation. Fig 17b showed that the $k_5$ for downed CWD also decreased exponentially with an increase in wood size, but the decrease in the $k_5$ for standing CWD with wood size increase was a power function.

## Discussion

### Differences in decomposition models

There were some differences in the calculated decomposition constants of CWD using different equations (i.e., Eqs 21–24). Since one of those equations (Eq 24) contains two constants ($k_4$

**Table 2. CWD decomposition constants calculated using different equations to assess model sensitivity to wood density and size\*.**

| Factors | DS | Downed CWD | | | | Standing CWD | | | |
|---|---|---|---|---|---|---|---|---|---|
| | | $k_1$ | $k_2$ | $k_3$ | $k_4$ | $k_1$ | $k_2$ | $k_3$ | $k_4$ |
| Wood Density (g cm$^{-3}$) | 0.3 | 0.1967 | 0.202 | 0.202 | 0.2324 | 0.1709 | 0.199 | 0.189 | 0.1882 |
| | 0.4 | 0.1938 | 0.201 | 0.200 | 0.2286 | 0.1592 | 0.195 | 0.181 | 0.1626 |
| | 0.5 | 0.1915 | 0.200 | 0.198 | 0.2233 | 0.1529 | 0.191 | 0.175 | 0.1467 |
| | 0.6 | 0.1892 | 0.199 | 0.196 | 0.2181 | 0.1478 | 0.189 | 0.171 | 0.1365 |
| | 0.7 | 0.1871 | 0.198 | 0.195 | 0.2159 | 0.1431 | 0.187 | 0.168 | 0.1266 |
| | 0.8 | 0.1849 | 0.198 | 0.194 | 0.211 | 0.1384 | 0.184 | 0.164 | 0.1198 |
| | 0.9 | 0.1826 | 0.197 | 0.192 | 0.2062 | 0.1332 | 0.182 | 0.160 | 0.1101 |
| | 1.0 | 0.1800 | 0.196 | 0.191 | 0.2023 | 0.1278 | 0.179 | 0.156 | 0.1021 |
| | Size | Softwood | | | | Hardwood | | | |
| Downed CWD (size) | 1 | 0.3180 | 0.281 | 0.308 | 0.3955 | 0.3578 | 0.270 | 0.325 | 0.4861 |
| | 2 | 0.2636 | 0.275 | 0.273 | 0.3140 | 0.2777 | 0.279 | 0.283 | 0.3521 |
| | 3 | 0.2219 | 0.243 | 0.235 | 0.2596 | 0.2358 | 0.255 | 0.248 | 0.2919 |
| | 4 | 0.1962 | 0.214 | 0.208 | 0.2255 | 0.2097 | 0.228 | 0.221 | 0.2606 |
| | 5 | 0.1792 | 0.196 | 0.190 | 0.2086 | 0.1917 | 0.209 | 0.203 | 0.2319 |
| | 6 | 0.1669 | 0.183 | 0.177 | 0.1889 | 0.1790 | 0.195 | 0.189 | 0.2137 |
| Standing CWD (size) | 1 | 0.3042 | 0.286 | 0.305 | 0.3685 | 0.3393 | 0.277 | 0.322 | 0.4363 |
| | 2 | 0.2374 | 0.278 | 0.263 | 0.2501 | 0.2441 | 0.286 | 0.272 | 0.2951 |
| | 3 | 0.1908 | 0.238 | 0.218 | 0.1736 | 0.1956 | 0.252 | 0.228 | 0.2114 |
| | 4 | 0.1657 | 0.209 | 0.191 | 0.1484 | 0.1694 | 0.221 | 0.199 | 0.1720 |
| | 5 | 0.1478 | 0.189 | 0.171 | 0.1256 | 0.1505 | 0.200 | 0.179 | 0.1502 |
| | 6 | 0.1357 | 0.175 | 0.158 | 0.1122 | 0.1382 | 0.186 | 0.165 | 0.1379 |

\* DS, density, g cm$^{-3}$; Size, wood size that is cm in diameter; the sizes of the classes 1 to 6 are 4.5–7.49, 7.5–14.99, 15.0–22.49, 22.5–29.99, 30.0–37.49, >37.5 cm in diameter, respectively. Results from simulation under the subtropical climatic conditions at Santee, South Carolina, USA.

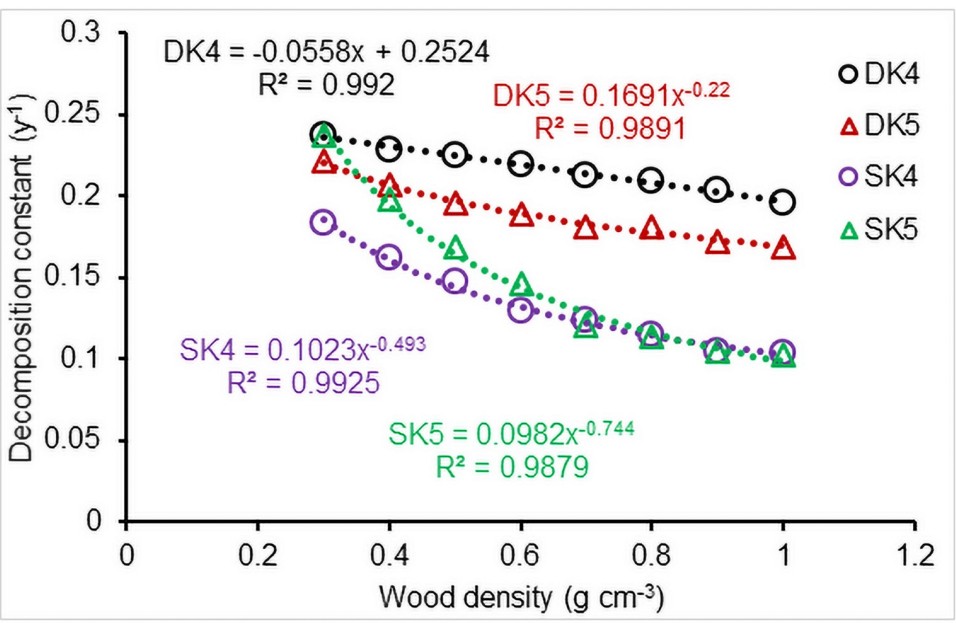

**Fig 16. CWD decomposition versus wood density.** Decomposition constants calculated using Eq 24; DK, downed CWD; SK, standing CWD. Results from simulation under the subtropical climatic conditions at Santee, South Carolina, USA.

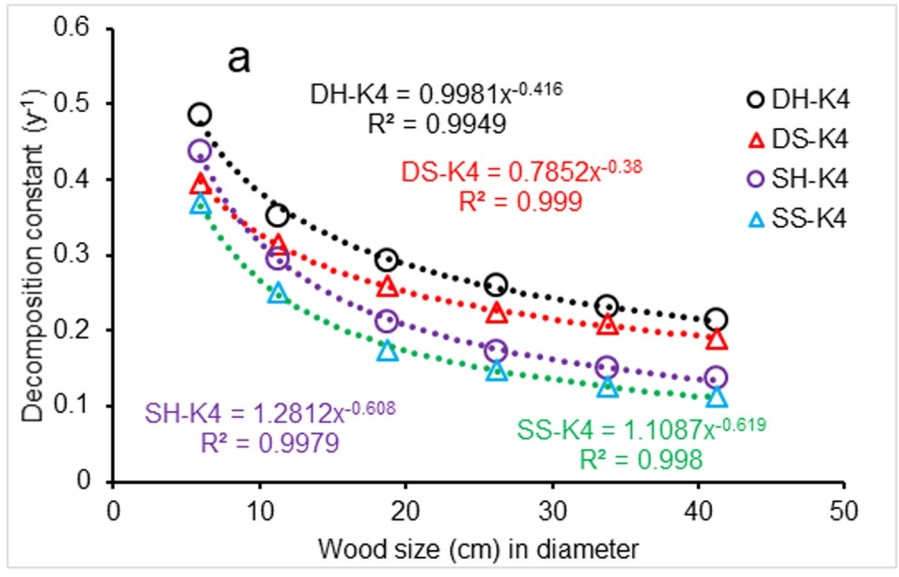

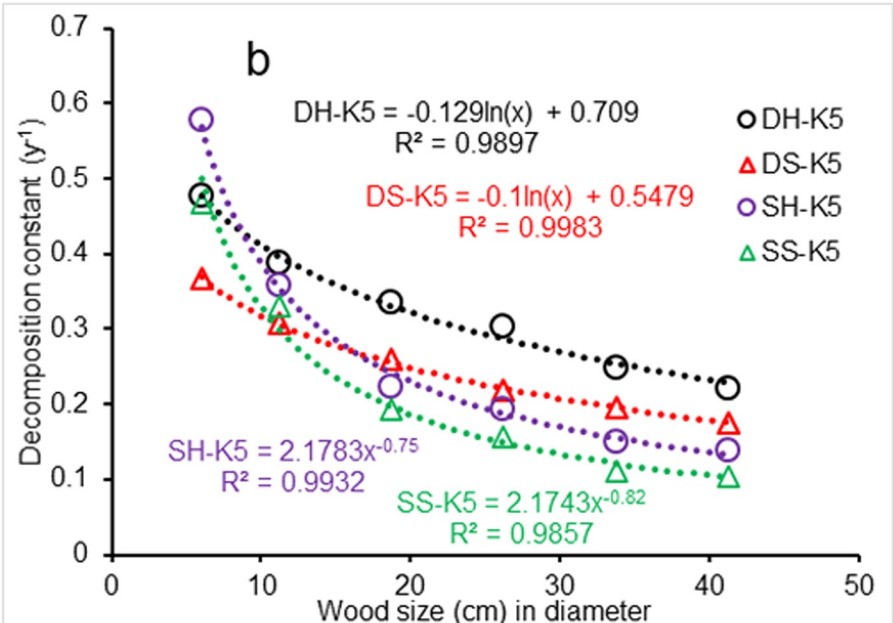

**Fig 17. Impact of wood size on the CWD decay.** DH, downed hardwood; DS, downed softwood; SH, standing hardwood; SS, standing softwood. Values for k4 (a) and k5 (b) were calculated using Eq 24. Results from simulation under the subtropical climatic conditions at Santee, South Carolina, USA.

and $k_5$), $T_{50}$ was used to compare the differences among the equations. The mean $T_{50}$ based on the decomposition constants from Eqs 21–24 for downed CWD in the eighty-nine sites were 11.7±10.9, 10.5±9.0, 10.8±9.8 and 12.6±9.8, respectively, and the means for standing CWD were 13.4±10.4, 11.2±8.8, 12.0±9.3 and 15.1±9.1, respectively. Although these means of $T_{50}$ based on different equations were seemingly similar, absolute $t$ values from paired sample t-test were between 3.0 and 19.7 for $T_{50}$ based on the decomposition constants from Eqs 21–24 for downed CWD, and between 11.3 and 58.4 for standing CWD decomposition; thus, these $t$

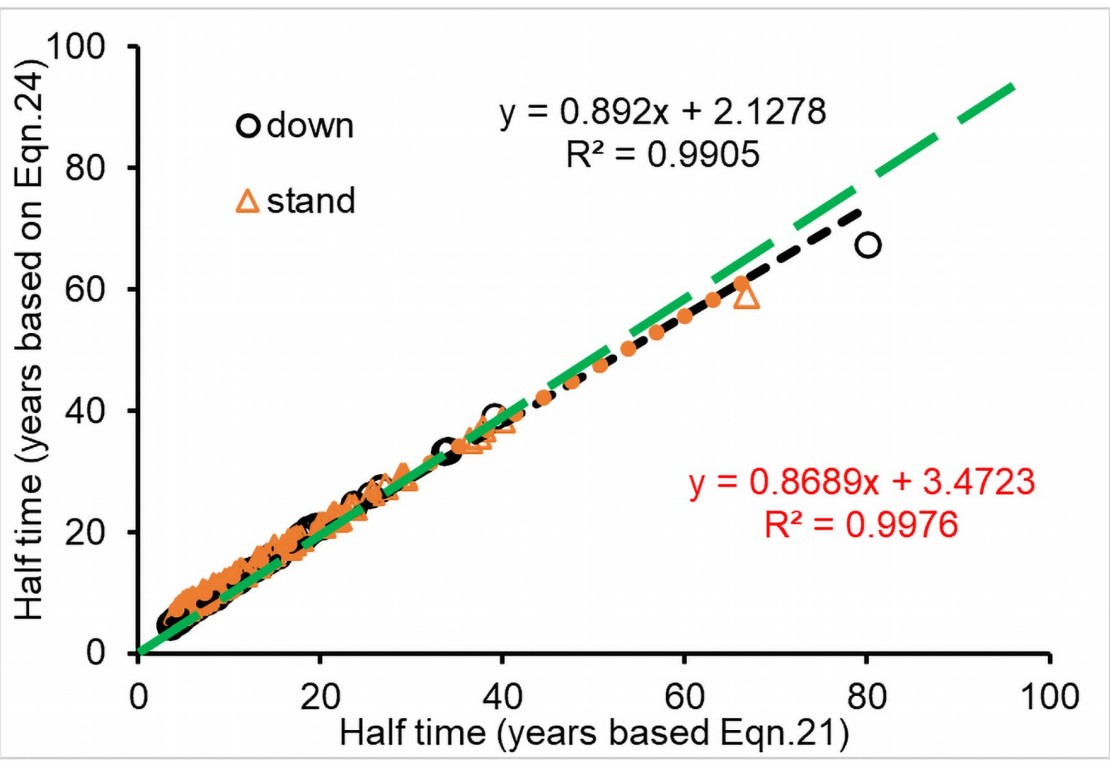

**Fig 18. Comparison of T$_{50}$ (years to fifty percent mass loss) calculated using decomposition constants from Eqs 21 and 24; green dashed line, 1:1.**

values were larger than the critical $t_c$ = 2.6, indicating that the calculated $T_{50}$ were different among decomposition models.

$T_{50}$ based on the $k$ from Eq 21 for both downed and standing CWD decomposition was significantly correlated to the $T_{50}$ based on Eq 24 ($n$ = 89, R$^2$>0.99, $P$<0.001; Fig 18). However, the intercept (3.47) for standing CWD decomposition was about 23% of the mean $T_{50}$ (15.1), and the slope was only 0.869. The mean $T_{50}$ based on Eq 21 was about 6.9% smaller than that based on Eq 24 for downed CWD decomposition, about 16.9% smaller than that for assessing standing CWD decomposition. The absolute error between two equations (Eqs 21 and 24) was 9.5 and 13.3% for the T$_{50}$ of downed and standing CWD decomposition, respectively. Similarly, the mean errors between $T_{50}$ based on Eq 24 and those based on Eqs 22 and 23 were 16.7 and 14.3% for downed CWD, respectively, and 26.0 and 20.8% for standing CWD, respectively. These metrics indicate that there are some differences among the decomposition models used to assess CWD decomposition rate, perhaps single exponential functions might function poorly for assessing CWD decomposition, especially for assessing standing CWD.

## Climatic factors

The sensitivity analysis for the model inputs showed that CWD decomposition was sensitive to temperature, precipitation and snowfall. The CWD decomposition rates increased non-linearly with temperature, with the rates increasing only slightly when annual mean temperature was over 22˚C and not increasing with mean temperature when temperature was over 25˚C. Wood-decomposers can only increase their activity with an increase in temperature when temperature is lower than their optimal survival temperature, and may decrease when

temperature is over the optimal temperature. For example, the optimal temperature for fungi is 20 to 30°C for most fungi and lower than 20°C for some fungi [75, 76].

When annual snowfall was over 200 kg m$^{-2}$, the CWD decomposition rate decreased little with decreasing temperature, and the rate might even slightly increase with snowfall increment due to greater thermal insulation from a deeper snowpack. The exponential increase in CWD decomposition rates with temperature from this model was consistent with the findings of Herrmann and Bauhus [77].

CWD decomposition was highly responsive to changes in precipitation, but there was a difference between downed and standing CWD. This difference can be related to wood moisture content in non-flooding environments. Standing CWD moisture content is mainly controlled by air humidity, while moisture in downed CWD can be partially regulated by soil moisture in the days without rain. Accordingly, the mean moisture of downed CWD can be higher than that of standing CWD, causing downed CWD to decompose faster, as a moister environment is more suitable for fungal growth, especially for soft-rot fungi. However, a water saturated environment is not suitable for most fungi.

## Spatiotemporal differences in CWD decomposition

The results of the sensitivity analysis of the CWD decomposition rate from the eighty-nine sites showed that there were substantial spatial differences, with rates ranging from low in the boreal zone to high in tropical areas. However, the latitudinal trends in decomposition rate were different between downed and standing CWD. The decomposition rate decreased with logarithmic increases in latitude for downed CWD and decreased linearly with latitude for standing CWD. This is due to multiple factors that impact CWD decomposition at any given latitude, including temperature, precipitation and altitude. The correlation of decomposition constants to key eco-environmental conditions for both downed and standing CWD can be described as

$$k = a_1 + a_2 \times T + a_3 \times P - a_4 \times LAT - a_5 \times ALT + a_6 \times Snow \qquad (35)$$

where $a_1 - a_6$ are coefficients; $T$, $P$, $LAT$, $ALT$ and $Snow$ are annual mean temperature (°C), precipitation (mm), latitude (°), altitude (m) and snowfall (kg m$^{-2}$), respectively ($F = 89.8$ and 38.1 for the $k_4$ from Eq 24 for downed and standing CWD, respectively; and $F = 73.2$ and 49.0 for the $k_5$ from Eq 24, respectively; $n = 89$, $P<0.001$). Eq 35 shows that the value of the CWD decomposition constant can increase with increments of both temperature and precipitation, and decreases with increments of latitude and altitude, which is consistent with the results reported by Zhang et al. [78]. However, Eq 35 gives a different correlation between the decomposition constant $k$ and snowfall, in which $k$ can increase with snowfall increase, in contrast to the regression equation in Fig 11 for which $k$ decreased with an increase in logarithmic snowfall. Multiple factors can cause different relationships between CWD decomposition constant $k$ and snowfall. The regression in Fig 11 used the data only from the 69 sites that received snow, while Eq 35 used all 89 sites. Fig 11 results also indicated that the decomposition constant $k$ did not decrease with snowfall increase when annual snowfall was over 200 kg m$^{-2}$, and instead the $k$ might slightly increase with snowfall increase when snowfall was more than 300 kg m$^{-2}$.

There also was a temporal effect on CWD decomposition. Annual downed CWD decomposition rates from Eq 20 for five sites in different climate zones (Fig 19) show that the CWD decomposition rates are not constant because time is needed for decomposers to colonize. The different intercepts in the figure indicate that there are substantial differences in the colonization of decomposers at these locations, the larger the intercept, the faster the wood

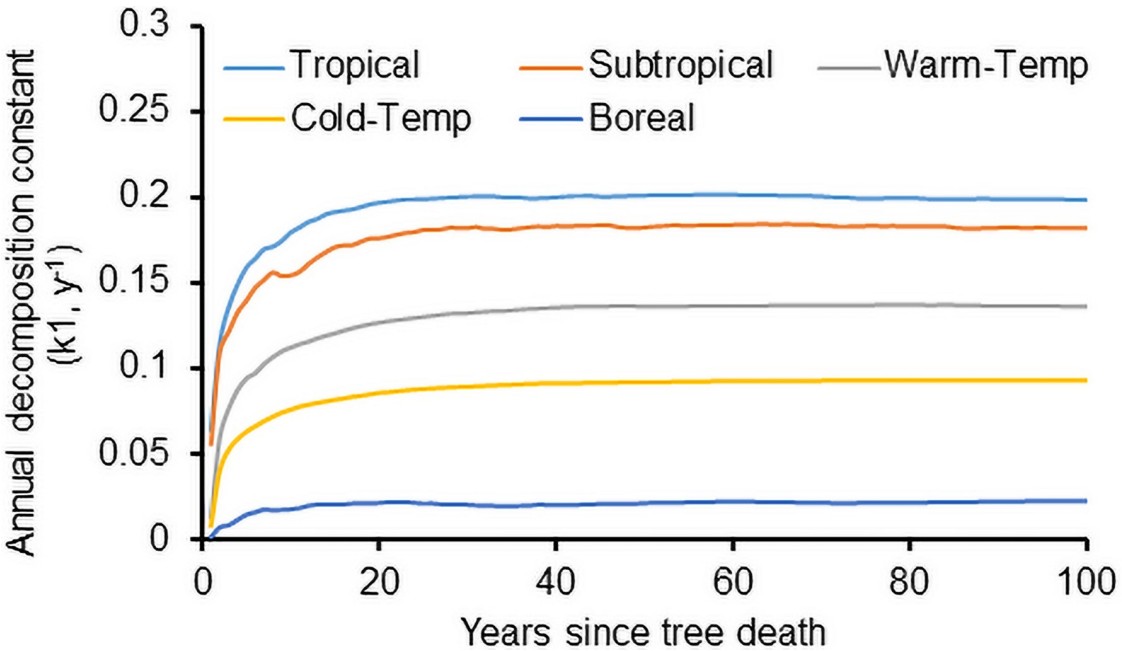

**Fig 19. Annual downed CWD decomposition rates at five sites in different climatic zones calculated using Eq 21; temp, temperate.**

colonization by wood-decomposers. The largest intercepts occurred at tropical and subtropical locations, because fungi colonize faster in these warm locations than cold sites, and termites often are present in warm locations.

### Impacts of wood properties and positions

The decomposition rate of both downed and standing CWD was sensitive to wood density (Table 2), with decomposition rate decreasing linearly ($R^2$>0.989, $n$ = 8, $P$<0.001) or via a power function ($R^2$>0.98, $P$<0.001) as density increased (Fig 16). Mackensen et al. [26] and Noh et al. [67] reported that wood moisture content had a negative linear correlation with wood density, which may be consistent with our results.

The decrease in the decomposition rate of downed CWD with an increase in wood density was less than that for standing CWD, which was related to the differences in wood moisture content and fungal activity between standing and downed CWD [43, 79]. The moisture of downed CWD is impacted by not only air humidity but also soil moisture. However, soil moisture has little influence on the moisture content of standing CWD.

The decomposition rates of both downed and standing CWD decreased non-linearly with the wood size increase ($P$<0.001), in agreement with Herrmann et al. [80]. However, our analysis showed a difference in the CWD decomposition rate between wood types, with hardwood decomposing faster than softwood, indicating that these decomposition rates are a function of wood biochemistry and the fungal and invertebrate wood-decomposers that colonize these different CWD species groups.

The decreasing trend in CWD decomposition with an increase in wood size should be related to changes in wood moisture content with time, because the time needed for water absorption by CWD and the time needed for evaporative water loss from the CWD are related to wood size. This decreasing trend in CWD decomposition should be also related to the ratio

of the available CWD surface to the wood mass, as the larger the ratio, the greater the probability of colonization by wood-decomposers.

## Contributions of different decomposers

There were significant differences in the contribution of wood-decomposer groups to CWD decomposition (S2 Fig), which were associated with geographical location and geomorphic features. Fungi are the main contributor to CWD decomposition everywhere, contributing over 50% of the total biological respiration. Termites occur only in warmer locations: tropical, subtropical and some warm temperate areas (purple bars in S2 Fig), and account for <40% of the total biological respiration based on our sensitivity analysis. The contribution of beetles with bacterial synergy to CWD decomposition is small (<10% of the total biological consumption; S2 Fig).

The fraction of termite respiration was inversely proportional to fungal respiration fraction, as termites quickly consumed the CWD reducing the mass of CWD available for fungal decomposition. The low fraction of termite respiration from standing CWD decomposition was due to minimal termite attack of standing CWD before they fall down.

## Comparison to results from the literature

We compiled published results from CWD decomposition studies, covering a wide geographic range (S4 and S5 Tables) [25, 26, 34, 81–109], spanning Asia, Europe, Oceania, North America and South America. Latitudes range from tropical Malaysia (1.4˚ N) [81] and Brazil (1.4˚ N) [82] to boreal Canada (56˚ N) [83], Russia (59˚ N) [84] and Sweden (68.3˚ N) [85] in the northern hemisphere, and from Australia [26] to New Zealand [86] and Brazil (2.5˚ S) [25] in the southern hemisphere. Most of these results are obtained from field studies, with additional information from data analysis using observations conducted by Zell et al. [34] and Herault et al. [87]. Because some decomposition rates were presented as decomposition constants and others were as $T_{50}$ and $T_{90}$ or $T_{95}$, we converted those decomposition constants to $T_{50}$ using Eq 18, and the global values of $T_{50}$ were presented in S4 Table.

The range in woody debris decomposition rates varied greatly, with $T_{50}$ ranging from 0.6 years ($k = 1.155$ y$^{-1}$) at a tropical plot (5˚18'N, 52˚55'W) in French Guiana [87] to 138.6 years ($k = 0.005$ y$^{-1}$) at a boreal site in the Leningrad region of Russia [88]. This range encompasses the results from our sensitivity analysis, which showed $T_{50}$ ranging from 4.4 years at a tropical location (15.8˚ N) to 67.3 years at a boreal site (65˚ N).

Our model results are within the global range of data found in the literature, because there are numerous factors influencing woody debris decomposition, including wood density and size, fresh CWD or snags, standing or fallen position, and dry or wet environmental conditions. Wood size is an important factor because wood debris with a small diameter decomposes faster than larger debris [88]. The average wood size for our sensitivity analysis was 26 cm in diameter. The mean wood density used for this sensitivity analysis from over eighty-nine sites was 0.55 g cm$^{-3}$, which is within wood densities used to assess woody debris decomposition in the literature.

Additionally, we did not have climate data for the tropical locations near the equator so we did not simulate woody debris decomposition at locations of latitude <14.2˚, and we did not consider a simulation for CWD decomposition in wetlands. However, our range of the woody debris decomposition rates was generally approximate to the global range ($T_{50}$: 2.1–88.9 years; $k$: 0.3289–0.0078 y$^{-1}$) reported by Zell et al. [34].

## Conclusions

Above-ground dead wood is a major carbon stock in forests. The process-based model CWDDAT has been developed to provide a tool to assess coarse wood decomposition and the associated linkages to the forest carbon cycle. Accordingly, the model provides a basis for simulating wood decomposition and the associated fluxes and fate of the wood carbon. While many aspects of wood decomposition are well documented in the literature, providing a sound basis for the design of this model, other aspects of the C cycle associated with wood decomposition have barely been studied. For example, there are very few reports about translocation of wood C into the forest floor and mineral soil, and the modes of wood-C movement into and through soil layers. Accordingly, CWDAT provides a useful framework for considering aspects of the forest C cycle that hasn't been feasible in other soil biogeochemical models. The development of CWDDAT has been predicated primarily on literature from the sub-tropical to sub-boreal zone in North America. Accordingly, further work may be required to better reflect the decomposer communities in tropical and boreal regions.

The model is designed to consider inherent site conditions, wood material, and the biological activity of decomposer communities over time. Sensitivity analysis showed that CWD decomposition was influenced by temperature, precipitation, snowfall, geographical location and geomorphic altitude, decomposer community composition, and wood properties, including density and size. The CWD decomposition rate increased substantially with an increase in temperature and precipitation, and decreased with an increment in latitude and altitude. Downed CWD decomposed faster than standing CWD. A deep snowpack can generate an insulating effect that decreases the effect of low temperatures on CWD decomposition.

The model effectively simulates the large-scale climatic (e.g., temperature and precipitation) effects on decomposer activity and the associated impacts on wood decomposition. An important feature of this model is the inclusion of termites as a major vector affecting wood decomposition. The simulated interaction between microbial communities and arthropods is temporally sensitive, reflecting differences in site conditions, wood properties and stage of decomposition.

## Supporting information

**S1 Table. Coordinates of eighty-nine sites used to obtain climate data for analyzing model sensitivity**[*]. [*]: Lat, latitude; Lon, longitude; all sites are located in North America; the climatic data were downloaded from Daymet database (Thornton et Al., 2016). Annual mean temperature, annual precipitation and elevation at each location are in S2 Table.
(DOCX)

**S2 Table. Altitude and climate at the sites used for model sensitivity analysis**[*]. [*] Ele, elevation; meanT, mean air temperature.
(DOCX)

**S3 Table. CWD decomposition constants calculated using different decomposition models for eighty-nine sites.** Values for $k_1$ calculated using Eqs 20 and 21 for downed and standing CWD, respectively; $k_2$, fitted without forcing the intercept; $k_3$ values fitted using forcing the intercept to 100% of the initial mass; and $k_4$ and $k_5$ values were based on Eq 24, respectively.
(DOCX)

**S4 Table. Published global decomposition constants for CWD**[*]. [*]:D, diameter range, cm; other information, including location and climate conditions are in S5 Table.
(DOCX)

**S5 Table. Locations and climate for the published global decomposition constants of CWD\***. \*: DM, decay model; SE, single exponent; NS, Northern and Southern Germany; LR, Leningrad Region; ML, multiple locations. The order of lines is the same as that in S4 Table.
(DOCX)

**S1 Fig. Time to fifty percent mass loss ($T_{50}$) of CWD decay calculated using different methods.** DK1 –DK3 (left) are calculated for downed deadwood decay using $k_1$, $k_2$ and $k_3$ from Eqs 21–23, respectively. DK4 (left) is calculated for the downed deadwood using $k_4$ and $k_5$ from the Eq 24. SK1 –SK4 (right) are for standing deadwood calculated using equations as described above for downed deadwood.
(TIF)

**S2 Fig. Respiratory contributions of different decomposers to deadwood decay at different sites.** The figure on the left for downed deadwood and that on the right is for standing deadwood.
(TIF)

# Acknowledgments

The authors wish to thank the technicians and scientists at the Santee Experimental Forest who have maintained the long-term records used for this analysis. We also thank Ben Bright, USDA Forest Service, Rocky Mountain Research Station, Moscow ID for producing the map. We acknowledge the useful comments of the reviewers of this manuscript.

# Author Contributions

**Conceptualization:** Zhaohua Dai, Carl C. Trettin, Martin F. Jurgensen, Brian T. Forschler.

**Data curation:** Zhaohua Dai, Deborah S. Page-Dumroese, Brian T. Forschler, Jonathan S. Schilling, Daniel L. Lindner.

**Formal analysis:** Deborah S. Page-Dumroese, Brian T. Forschler, Daniel L. Lindner.

**Funding acquisition:** Carl C. Trettin, Andrew J. Burton, Martin F. Jurgensen, Jonathan S. Schilling.

**Investigation:** Carl C. Trettin, Andrew J. Burton, Deborah S. Page-Dumroese, Jonathan S. Schilling, Daniel L. Lindner.

**Methodology:** Zhaohua Dai, Andrew J. Burton, Martin F. Jurgensen, Deborah S. Page-Dumroese, Brian T. Forschler, Jonathan S. Schilling, Daniel L. Lindner.

**Project administration:** Carl C. Trettin.

**Software:** Zhaohua Dai.

**Supervision:** Carl C. Trettin, Martin F. Jurgensen.

**Validation:** Zhaohua Dai.

**Writing – original draft:** Zhaohua Dai.

**Writing – review & editing:** Carl C. Trettin, Andrew J. Burton, Martin F. Jurgensen, Deborah S. Page-Dumroese, Brian T. Forschler, Jonathan S. Schilling.

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
