## [Decision Letter · Decision Letter 0]

6 Mar 2021

PONE-D-21-02698

Coarse Woody Debris Decomposition Assessment Tool: Model Development and Sensitivity Analysis

PLOS ONE

Dear Dr. Dai,

Thank you for submitting your manuscript to PLOS ONE. After careful consideration, we feel that it has merit but does not fully meet PLOS ONE’s publication criteria as it currently stands. Therefore, we invite you to submit a revised version of the manuscript that addresses the points raised during the review process.

We look forward to receiving your revised manuscript.

Kind regards,

Dafeng Hui, Ph.D.

Academic Editor

PLOS ONE

Journal Requirements:

2.In your Data Availability statement, you have not specified where the minimal data set underlying the results described in your manuscript can be found. PLOS defines a study's minimal data set as the underlying data used to reach the conclusions drawn in the manuscript and any additional data required to replicate the reported study findings in their entirety. All PLOS journals require that the minimal data set be made fully available. For more information about our data policy, please see http://journals.plos.org/plosone/s/data-availability.

4.Thank you for stating the following in the Acknowledgments Section of your manuscript:

"Funding to install the FWDE was provided by the USDA Forest Service and Michigan

Technological Univ. Current work is being supported by the U.S Dept. of Energy (DE764

SC0016235), US NSF (DEB 1754603) and the USDA Forest Service."

 "The funding sources have been listed in Acknowledgments."

"NO"

6.We note that Figure(s) 2 in your submission contain map images which may be copyrighted. All PLOS content is published under the Creative Commons Attribution License (CC BY 4.0), which means that the manuscript, images, and Supporting Information files will be freely available online, and any third party is permitted to access, download, copy, distribute, and use these materials in any way, even commercially, with proper attribution. For these reasons, we cannot publish previously copyrighted maps or satellite images created using proprietary data, such as Google software (Google Maps, Street View, and Earth). For more information, see our copyright guidelines: http://journals.plos.org/plosone/s/licenses-and-copyright.

a) You may seek permission from the original copyright holder of Figure(s) 2 to publish the content specifically under the CC BY 4.0 license. 

Additional Editor Comments:

I now have one report from an expert reviewer who considers this study interesting, but also raises several technique concerns, including logic in the introduction and some issues related to modeling. My decision is Major revision.

Reviewers' comments:

Reviewer's Responses to Questions

**Comments to the Author**

1. Is the manuscript technically sound, and do the data support the conclusions?

Reviewer #1: Partly

2. Has the statistical analysis been performed appropriately and rigorously? 

Reviewer #1: Yes

3. Have the authors made all data underlying the findings in their manuscript fully available?

Reviewer #1: No

4. Is the manuscript presented in an intelligible fashion and written in standard English?

Reviewer #1: No

5. Review Comments to the Author

Reviewer #1: This is potentially interesting study dealing with modelling yet poorly studied decomposition dynamics of coarse woody debris with qualifying factors influencing wood decomposition rate.

However, the manuscript needs major revisions.

The introduction would be more logical if the text would be focused on the factors influencing the decomposition rates and the mechanisms of this influence. The essence of the developed model should be better explained. Moreover, the objective of this study should be clearly stated.

The importance of processes leading to mass loss of CWD, such as fragmentation, can be stated in the introduction, not in the methods section. The same concerns bacteria and other biological agents. It is stated that the contribution of bacteria to overall deadwood decomposition is small, but later the equation including some figure for the mass of bacteria is presented. How this mass was estimated?

I did not understand, where the coefficients used in the separate model equations came from? Please, explain, how did you estimate e.g. fungal and beetle biomass, or the daily CWD consumed by termites. This information supported by references can be provided e.g. in supplements. It is impossible to evaluate the results of the sensitivity analysis without understanding model inputs. Moreover, the comparisons of decay rates calculated in different ways based on suggested model are of restricted value for potential users, because it is unclear, how where the decay constants calculated?

The next question is: how the environmental variables were included in the model? This is especially important as the main result from the sensitivity analysis is that deadwood decomposition was influenced by the main inputs, temperature, precipitation, snow, geographical location and geomorphic altitude, as well as wood properties such as density and size. How were these variables used as model inputs?

Conclusions should not repeat results!

The number of tables and figures can be shortened, some of them can be moved to supplements.

The terminology used should be consistent. For example, the terms coarse woody debris, woody debris, deadwood, dead wood are used throughout the text.

Fig. 1 could be better developed by adding e.g. references to the equations used, explaining, what the arrows and question marks mean etc. Fig. 1 contains mistakes: leaching

Here are some specific comments on the Introduction and methods showing that the text should be revised.

Abstract.

Lines 20-22. It would be better to use the same order: either from tropical to boreal or the other way round in listing temperature, precipitation and other ranges.

Line 22. Annual snowfall?

The terms ‘decomposition’ and ‘decay’ should be defined and should not be used as synonyms.

Introduction

Line 42. Habitats

Line 43. Potential risk of wildfires

Line 48. Remove the word ‘natural’.

Lines 55-56. The statement needs a reference.

Lines 61-62. ‘Rots’ do not ‘decompose’. (wording)

Lines 67-71. This is not true. There is a huge number of studies on the decomposition of CWD! The whole paragraph should be re-written.

Line 73. What is ‘wood location’?

Lines 76-77. What is ‘mixed with linear and exponential models’?

Lines 83-90. The point here is unclear. The essence of the models should be better described.

Methods

Line 109. The sentence is very unclear. The information from ‘the literature on CWD decomposition’ that was used in the model should be specified.

Line 177. Beetles are not wood decomposing

Line 304. The ‘decay constants’ cannot be ‘obtained from methods’.

6. PLOS authors have the option to publish the peer review history of their article (what does this mean?). If published, this will include your full peer review and any attached files.

Reviewer #1: **Yes: **Ekaterina Shorohova

---

## [Author Response · Author response to Decision Letter 0]

21 Apr 2021

Dear Dr. Dafeng Hui:

Thank you very much for your great work. All responses are in the separate file --Response_To_Editor_Reviewers-MS1.docx.

Response to the following important comments:

Response: while revising we have taken care to response to each point of reviewers' comments.

Response: Great. We uploaded as you suggested.

Response: Thank you for the kind reminder. We uploaded the untracked version as manuscript.

Thanks again.

Best Regards

Zhaohua

---

## [Decision Letter · Decision Letter 1]

5 May 2021

Coarse Woody Debris Decomposition Assessment Tool: Model Development and Sensitivity Analysis

PONE-D-21-02698R1

Dear Dr. Dai,

We’re pleased to inform you that your manuscript has been judged scientifically suitable for publication and will be formally accepted for publication once it meets all outstanding technical requirements.

Kind regards,

Dafeng Hui, Ph.D.

Academic Editor

PLOS ONE

Additional Editor Comments (optional):

Reviewers' comments:

Reviewer's Responses to Questions

**Comments to the Author**

1. If the authors have adequately addressed your comments raised in a previous round of review and you feel that this manuscript is now acceptable for publication, you may indicate that here to bypass the “Comments to the Author” section, enter your conflict of interest statement in the “Confidential to Editor” section, and submit your "Accept" recommendation.

Reviewer #1: All comments have been addressed

2. Is the manuscript technically sound, and do the data support the conclusions?

Reviewer #1: Yes

3. Has the statistical analysis been performed appropriately and rigorously? 

Reviewer #1: Yes

4. Have the authors made all data underlying the findings in their manuscript fully available?

Reviewer #1: Yes

5. Is the manuscript presented in an intelligible fashion and written in standard English?

Reviewer #1: Yes

6. Review Comments to the Author

Reviewer #1: (No Response)

7. PLOS authors have the option to publish the peer review history of their article (what does this mean?). If published, this will include your full peer review and any attached files.

Reviewer #1: **Yes: **Ekaterina Shorohova

---

## [Editor Report · Acceptance letter]

18 May 2021

PONE-D-21-02698R1 

Coarse Woody Debris Decomposition Assessment Tool: Model Development and Sensitivity Analysis 

Dear Dr. Dai:

I'm pleased to inform you that your manuscript has been deemed suitable for publication in PLOS ONE. Congratulations! Your manuscript is now with our production department. 

Kind regards, 

on behalf of

Dr. Dafeng Hui 

Academic Editor

PLOS ONE